

# The motion of trees in the wind: a data synthesis

Toby D. Jackson[1], Sarab Sethi[2], Ebba Dellwik[3], Nikolas Angleou[3], Amanda Bunce[4], Tim van Emmerik[5], Marine Duperat[6], Jean-Claude Ruel[6], Axel Wellpott[7], Skip Van Bloem[8], Alexis Achim[9], Brian Kane[10], Dominick M. Ciruzzi[11], Steven P. Loheide II[11], Ken James[12], Daniel Burcham[13], John Moore[14], Dirk Schindler[15], Sven Kolbe[15], Kilian Wiegmann[16], Mark Rudnicki[17], Victor J. Lieffers[18], John Selker[19], Andrew V. Gougherty[20], Tim Newson[21], Andrew Koeser[22, 23], Jason Miesbauer[24], Roger Samelson[25], Jim Wagner[26], David Coomes[1], Barry Gardiner[27]

[1] Plant Sciences, University of Cambridge, CB2 3EA, UK
[2] Department of Mathematics, Imperial College London, UK
[3] Department of Wind Energy, Technical University of Denmark, Frederiksborgvej 399, Roskilde, 4000, Denmark
[4] Department of Natural Resources, University of Connecticut, Mansfield, CT 06269, USA
[5] Hydrology and Quantitative Water Management Group, Wageningen University, Wageningen, The Netherlands
[6] Department of Wood and Forest Sciences, Laval University, Quebec, G1V 0A6, Canada
[7] Bavarian State Institute of Forestry (LWF), Hans-Carl-von-Carlowitz-Platz 1, D-85354 Freising
[8] Baruch Institute of Coastal Ecology and Forest Science, Clemson University, PO Box 596, Georgetown, SC 29442, USA
[9] Centre de recherche sur les matériaux renouvelables, Département des sciences du bois et de la forêt, Université Laval, Québec, QC G1V 0A6, Canada
[10] Department of Environmental Conservation, University of Massachusetts, Amherst, MA 01003, USA
[11] Civil and Environmental Engineering, University of Wisconsin Madison, Madison, WI
[12] School of Ecosystem and Forest Sciences, Faculty of Science, University of Melbourne, Melbourne, Australia
[13] Centre for Urban Greenery and Ecology, National Parks Board, 259569 Singapore
[14] Timberlands Ltd., Rotorua 3010, New Zealand
[15] Environmental Meteorology, University of Freiburg, Germany
[16] Argus Electronics gmbh, Erich-Schlesinger-Str. 49d, 18059 Rostock
[17] College of Forest Resources and Environmental Science, Michigan Technological University, Houghton, MI 49931 USA
[18] Renewable Resources Dept, University of Alberta, USA
[19] Oregon State University, Corvallis, OR 97331, USA
[20] Department of Botany, University of British Columbia, Canada
[21] Department of Civil and Environmental Engineering, Western University, Canada
[22] Department of Environmental Horticulture, IFAS, University of Florida
[23] Gulf Coast Research and Education Center, 14625 County Road 672, Wimauma, FL 33598, United States
[24] The Morton Arboretum, Lisle, IL 60532, USA
[25] College of Earth, Ocean, and Atmospheric Sciences, Oregon State University, Corvallis OR 97331 USA
[26] Oregon Research Electronics, Tangent, OR 97389, USA
[27] Institut Européen de la Forêt Cultivée, 69 route d'Arcachon, 33612, Cestas, France

*Correspondence to*: Toby Jackson (tobydjackson@gmail.com)



**Abstract.**

1. Interactions between wind and trees control energy exchanges between the atmosphere and forest canopies. This energy exchange can lead to the widespread damage of trees and wind is a key disturbance agent in many of the world's forests.

However, most research on this topic has focused on conifer plantations, where risk management is economically important, rather than broadleaf forests, which dominate the forest carbon cycle. This study brings together all available tree motion time-series data to systematically evaluate the factors influencing tree responses to wind loading, including data from both broadleaf and coniferous trees in forests and open environments.

2. We found that the two most descriptive features of tree motion were: (a) the fundamental frequency, which is a measure of the speed at which a tree sways and is strongly related to tree height, and (b) the slope of the power spectrum, which is related to the efficiency of energy transfer from wind to trees. Intriguingly, the slope of the power spectrum was found to remain constant from medium to high wind speeds for all trees in this study. This suggests that, contrary to some predictions, damping or amplification mechanisms do not change dramatically at high wind speeds and therefore wind damage risk is

related, relatively simply, to wind speed.

3. Conifers from forests were distinct from broadleaves in terms of their response to wind loading. Specifically, the fundamental frequency of forest conifers was related to their size according to the cantilever beam model (i.e. vertically distributed mass), whereas broadleaves were better approximated by the simple pendulum model (i.e. dominated by the

crown). Forest conifers also had a steeper slope of the power spectrum. We interpret these finding as being strongly related to tree architecture, i.e. conifers generally have a simple shape due to their apical dominance, whereas broadleaves exhibit a much wider range of architectures with more dominant crowns.

## 1. Introduction

Tree size and growth rate are influenced by their local wind environment (Bonnesoeur et al., 2016; MacFarlane and Kane,

2017) which in turn impact on forest carbon storage and dynamics. Monitoring the motion of trees in the wind can help us understand this interaction and model the risk of wind damage (Moore et al., 2018), which is a key driver of the forest carbon cycle in both temperate and tropical regions (Espírito-Santo et al., 2014; Schelhaas et al., 2003; Senf and Seidl, 2020).

Trees have characteristic and recognisable swaying patterns. This means that tree motion time-series are distinct from each other and from the time-series of local wind speeds. These tree motion characteristics are determined by tree size, shape and,

to a lesser extent, material properties (Dargahi et al., 2020; Jackson et al., 2019; Sellier and Fourcaud, 2009). Therefore, just as some types of trees have recognisable architectures, we expect them to have distinctive patterns of motion in response to wind loading. For example, trees in dense forests generally have a slender form with a small crown near the top (MacFarlane

and Kane, 2017), which leads to a slow, pendulum-like motion (Sellier and Fourcaud, 2009).

Previous data syntheses have focused on the fundamental sway frequency ($f_0$) of conifers and have found that larger, heavier trees sway more slowly than shorter, lighter ones (Moore and Maguire, 2004). This finding demonstrates that conifers can be approximated by a cantilever beam (i.e. a beam with distributed mass), but it is unclear whether this model extends to

other types of trees. Additionally, a tree's $f_0$ has been observed to change over time in response to variations in tree mass and elasticity. These changes can be used as a proxy to measure phenology (Bunce et al., 2019; Gougherty et al., 2018) and water status of trees (Ciruzzi and Loheide, 2019) at high time resolution (e.g. every 10 minutes or hour) using low-cost sensors. Therefore, tree motion characteristics are expected to change with the seasons, with the presence or absence of leaves in deciduous trees and the freezing soil in high latitudes having important effects (Bunce et al., 2019).


Energy is transferred from the wind to a tree due to aerodynamic drag and is then dissipated by damping mechanisms in the tree. This balance determines a tree's risk of wind damage. A number of processes, both amplifying and damping the tree motion, have been suggested to become significant in the high wind speed regime. For example, damping by branching, whereby the branching patterns of open-grown trees increase damping efficiency by transferring energy to the outermost

branches where it can be efficiently dissipated (de Langre, 2008; De Langre, 2019; Spatz and Theckes, 2013; Theckes et al., 2011). Another example is the dynamic amplification of the tree motion as the peak energy of the wind spectrum approaches the natural frequency of the tree, inducing resonant effects (BLACKBURN et al., 1988; Ciftci et al., 2013; Holbo et al., 1980; Oliver and Mayhead, 1974; Rodriguez et al., 2008). We cannot measure these processes in the field, but the slope of the power spectrum ($S_{freq}$) can be used as an overall measure of energy transfer between wind and tree at different frequency

ranges (van Emmerik et al., 2018; Van Emmerik et al., 2017). We therefore calculate $S_{freq}$ for each hour of tree motion data and look for changes with increasing wind speed.

This study brings together all available data on tree motion from tropical, temperate and boreal regions to explore key similarities and differences in the motion of these trees in the wind. We do this by comparing features calculated from the

tree motion time series data. While $S_{freq}$ and $f_0$ are clearly important features of tree motion, thousands of other time-series features have been developed across multiple disciplines which may provide further insight. Fulcher and Jones, (2017) compiled a set of over 7,700 time-series features and (Lubba et al., 2019) distilled them to the most descriptive and minimally redundant 22 features ('*catch22*' features). We use these *catch22* features as well as $f_0$ and $S_{freq}$ (and associated features described in the methods) to systematically explore the similarities and differences in the way trees move. We



expect to find differences between conifers and broadleaves due to their different architectures (Jackson et al., 2019) as well as between open-grown trees and those in forests.

The specific questions we address are:

Q1. What models best predict the fundamental frequency of broadleaf trees and conifers?

Q2. Are characteristics of tree motion distinct between tree groups (conifers, broadleaves, forest, open-grown)?

Q3. To what extent does increasing wind speed (or the change between summer and winter) change the characteristics of tree motion?

## 2. Methods

### 2.1 Description of the data

We collated data from 20 studies that included 238 trees and more than 1 million hours of tree motion data at resolutions
ranging from 4 to 20 Hz. These studies used three types of sensor: (1) strain gauges which measure the bending strains at the base of the tree (Moore et al., 2005) (2) inclinometers measuring the changing inclination angle of the trunk (Bunce et al., 2019; Schindler et al., 2010) and (3) accelerometers measuring the acceleration at the top of the trunk (Van Emmerik et al., 2017). We refer to these in general as measures of tree deflection. Five studies additionally measured branch motion, which is not used here but may provide further insight in future studies. Some datasets span multiple years, while others are
confined to a short time during windy conditions. Most data sets contained tree height ($H$) and diameter at breast height ($dbh$) measurements and tree species data. Many of the individual studies in our data set did not measure high-resolution (>1 Hz) wind speeds because this requires sonic anemometers and so substantially increases the cost of the field study. Also, in most cases the wind measurements were not in the same location as the trees, which hinders detailed analysis at high time-resolution. We therefore focus on analyses which do not require these data (although we explore this data in supplementary
S2). For the purposes of our analyses, we categorized trees as broadleaf or conifer and as open-grown or forest (as defined by the data owners). Table 1 gives an overview of the data sets.

### 2.2 Data processing and feature extraction

We re-sampled each time-series to the lowest sampling frequency in the data set (4 Hz) to avoid sampling rate confounding our analysis. This frequency was sufficient to capture tree motion characteristics for all trees in our sample.
Although each type of sensor measures a different property of tree motion, these properties are very strongly correlated and are complementary measures of tree motion. Each type of sensor has its own units, so absolute values are only comparable within sites. All sensors are susceptible to a drift in the measurements resulting in an offset which varies slowly with temperature and other environmental factors. This is particularly evident in the strain gauges, presumably because they are directly attached to the tree and so respond to the daily changes in tree diameter. We removed this offset using a 10-minute
high-pass Butterworth filter and applied the same filter to inclinometer and accelerometer data to ensure a fair comparison.



This filter effectively removes the low-frequency (variations slower than 10 minutes) part of the tree motion, including any mean displacement and offset in the data. This period seems reasonable to capture most effects of wind-tree interaction, although open-grown trees exposed to strong winds may experience slowly varying displacements due to the mean wind speed on this timescale (Angelou et al., 2019; James et al., 2006).


| Site | Country | Year | Group | # Trees | Sensor | Resolution (Hz) | Wind resolution |
|---|---|---|---|---|---|---|---|
| Rivox | UK | 1988 | Conifer Forest | 4 | Strain gauges | 10 | 10 Hz |
| Kershope | UK | 1989 | Conifer Forest | 11 | Strain gauges | 10 | 10 Hz |
| Whitecourt | Canada | 1999 | Conifer Forest | 10 | Inclinometers | 10 | 0.2 Hz |
| Clocaenog | UK | 2005 | Conifer Forest | 9 | Strain gauges | 4 | 10 Hz |
| Kyloe | UK | 2006 | Conifer Forest | 9 | Strain gauges | 4 | 10 Hz |
| Belchertown | USA, MA | 2006 | Broadleaf Open | 9 | Strain gauges | 20 | - |
| Various | Australia | 2006 | Mixed Open | 9 | Strain gauges | 20 | 1 Hz |
| Guanica | Puerto Rico | 2007 | Broadleaf Forest | 9 | Strain gauges | 4 | |
| - | USA, OR | 2011 | Broadleaf Open | 3 | Accelerometer | 20 | - |
| Storrs | USA, CT | 2013 | Broadleaf Forest | 13 | Inclinometer | 10 | 30 min |
| Alexandra | Singapore | 2013 | Broadleaf Open | 4 | Accelerometer | 10 | - |
| Orange | USA, CT | 2015 | Broadleaf Forest | 14 | Inclinometer | 10 | 30 min |
| Torrington | USA, CT | 2015 | Mixed Forest | 14 | Inclinometer | 10 | 30 min |
| Manaus | Brazil | 2015 | Broadleaf Forest | 19 | Accelerometer | 10 | 15 min |
| Various | USA | 2016 | Broadleaf Open | 9 | Accelerometer | 10 | - |
| Hartheim | Freiburg | 2016 | Conifer Forest | 4 | Inclinometers | 10 | 10 Hz |
| Morton arboretum | USA, IL | 2016 | Broadleaf Open | 8 | Accelerometers | 10 | |
| Wytham Woods | UK | 2016 | Broadleaf Forest | 21 | Strain gauges | 4 | 1 Hz |
| Riso | Denmark | 2017 | Broadleaf Open | 1 | Strain gauges | 20 | 20 Hz |
| Danum | Malaysia | 2017 | Broadleaf Forest | 19 | Strain gauges | 4 | 1 Hz |
| Fond du Lac | USA, WI | 2018 | Broadleaf Open | 4 | Accelerometer | 16 | 5 min |
| Trout Lake | USA, WI | 2018 | Broadleaf Forest | 8 | Accelerometer | 16 | 1 min |
| Montmorency | Canada | 2018 | Conifer Forest | 15 | Strain gauges | 5 | 5 Hz |
| Big Sur | USA, CA | 2018 | Conifer Forest | 2 | Accelerometer | 5 | 10 min |
| Appalachian Lab | USA, MD | 2019 | Broadleaf Open | 20 | Accelerometer | 10 | - |
| Various | Germany | 2019 | Mixed Open | 16 | Inclinometer | 20 | - |
| Freiburg | Germany | 2019 | Conifer Open | 1 | Accelerometer | 10 | 10 Hz |
| Hartheim | Germany | 2019 | Conifer Open | 1 | Accelerometer | 10 | 10 Hz |

**Table 1 - Overview of the data sets used in this study. Links to the data sets are provided in the online version.**



We selected 1-hour samples during windy conditions and calculated all tree motion features for all 238 trees in our data set.

We first calculated features from the combined horizontal components of tree motion. We fitted an extreme value distribution and retained the shape parameter, k, which does not depend on the absolute values and so is comparable across sites. The scale and location parameters of the extreme value distribution depend on absolute values and we did not use them in this study. Next, we calculated features from tree motion along a single axis of motion, since combining the two horizontal axes distorts important features such as $f_0$. The coordinate systems of each tree were not aligned with the wind

direction. We therefore used a principal component analysis to select the axis with highest variance, the 'stream-wise' component, and the perpendicular 'cross-stream' component. We calculated the ratio of the explained variance in the two components, a measure of how elliptical the tree sway pattern is. We found that the cross-stream component did not contribute additional information and so all analysis in this study used the stream-wise component of tree sway only. This is similar to using a time series from a single, horizontal axes of motion which is aligned with the dominant direction of the

wind loading.

We centered and scaled the stream-wise component of tree sway and calculated the *catch22* features (Lubba et al., 2019). These 22 features were selected to be maximally descriptive over a wide range of time-series data. Examples of these features are: (1) the time interval between successive extreme events above the mean (2) the mean error from forecasting the

next value of the time-series as the mean of the previous three and (3) the centroid of the Fourier power spectrum. For more details of these features please see the supplementary materials and the original publications (Fulcher and Jones, 2017; Lubba et al., 2019).

We also calculated the power spectral density of the stream-wise component using Welch's method (Welch, 1967). We then

calculated the slope of the power spectrum, $S_{freq}$, by fitting a linear model between log-transformed frequency and log-transformed power spectral density (Van Emmerik et al., 2017). Since different frequency ranges represent different physical scales of energy transfer between wind and trees (see Figure 1), we calculated this slope for multiple frequency ranges (0.05-0.8 Hz, 0.8-2 Hz and 0.05-2 Hz). We test the sensitivity of this feature to different frequency ranges and fitting methods in the supplementary materials (S1) and found that the trends and differences between trees are not sensitive to the choice of

frequency range, although the absolute values are. Finally, we extracted the frequency, width, height and dominance of the $f_0$ peak from the stream-wise component. The dominance was defined as the ratio of the fundamental peak height to the sum of all the heights of all other peaks in the spectrum (Jackson et al., 2019). In this study we focus on the features which help distinguish between tree types or are correlated with tree size. We do not discuss those features which don't meet these criteria, although these features may also exhibit interesting trends that warrant further study.






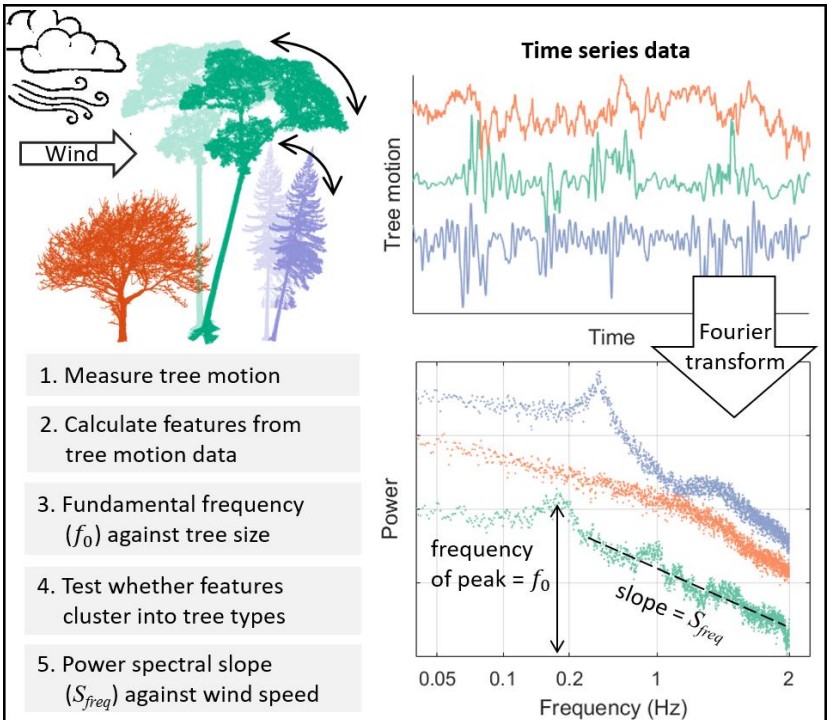

**Figure 1 – Overview of the study design. Top left – conceptual diagram of data collection showing three 'types' of trees (colours) swaying in the wind. Top right – five-minute sample of tree motion data for three trees. Bottom right – Fourier transformed data used to calculate the fundamental frequency, slope of the power spectrum and related features.**

## 2.3 Linear models of fundamental frequency against tree size and tree type

The aim of this analysis was to test whether the previously observed relationship between $f_0$ and tree size hold across the additional data in this study, especially the broadleaves. In addition to the detailed tree motion data for the 238 trees described above, we collated summary data on $f_0$ and tree size for a further 591 trees (Baker, 1997; Moore and Maguire, 2004). We used a linear regression to explore the relationship between $f_0$ and tree size. We tested the correlation with a simple pendulum model $\sqrt{\frac{1}{H}}$, and a cantilever beam model $\frac{dbh}{H^2}$. Initial results showed that different measures of tree size models fit better for conifers and broadleaves, so we performed a separate analysis for each tree type instead of using multiple linear regression. Additionally, we tested each model after log-transforming both variables and report both results. We report the best fit models based on the coefficient of determination ($R^2$) and Akaike's Information Criterion (AIC) (Pan, 2001).

## 2.4 Classifying tree types based on tree motion features

The aim in this analysis was to test whether different types of trees exhibited distinct, characteristic motion in the wind. We first tested whether the trees grouped into the predefined groups (open-grown broadleaf, forest broadleaf, open-grown

conifers, forest conifer) using a principal component analysis. We then used supervised methods (linear discriminant analysis and multinomial regression) to determine which two features best classified the trees into types. We also used a 10-fold cross validation with a 50:50 training to testing data ratio to test whether the trees could be classified by type according to the features of their motion using a random forest model. We assessed the classification accuracy using Cohens kappa (Cohen, 1960). We centered and scaled all features prior to analysis.

Note that our sample did not contain enough open-grown conifers to perform statistical analysis on this group. We therefore performed the above analysis on only 86 forest broadleaves and the 62 open-grown broadleaves and the 63 forest conifers. The 12 open-grown conifers were projected onto the resulting axes in Figure 3. Since the sample sizes in each group were not equal, we randomly re-sampled the larger groups to the size of the smallest (N=62) and repeated each analysis with this approach 50 times. We use this evenly sampled data set in the linear discriminant analysis and random forest classification analysis, but not in the principal component analysis because it did not make a substantial difference.

We tested whether the classification analysis was likely impacted by the unequal tree size distributions in each tree type category. We used a subset of 168 trees (86 forest broadleaves, 54 forest conifers and 28 open-grown broadleaves) for which we have raw tree motion data as well as tree size data. We used a mixed-effects model (*lme4*) to predict tree size from the tree motion features and tree type (included in the model as a factor). We conducted separate analysis for tree height and *dbh* and the best predictors were ordered using a two-way stepwise procedure based on minimizing AIC (Pan, 2001). The factor "tree type" was the 9[th] most explanatory feature in the model of height and 6[th] in the model of *dbh*. This demonstrates that tree size is more strongly related to tree motion features than it is to tree type. Therefore, the relationship between tree type and tree motion features is unlikely to be confounded by differences in tree size, and hence the results of our classification analyses are valid..

## 2.5 Tracking changes in key tree motion characteristics with increasing wind speed

The aim of this analysis was to test whether the relationship between tree motion and wind speed varies between summer and winter, and whether $S_{freq}$ changes in a predictable way with wind speed. For the data sets with long-term tree motion and wind speed data (N=103 trees, 74 forest broadleaves and 29 forest conifers) we calculated each of the features described above for every hour of data available. We also calculated the hourly mean wind speed, the maxima and 99th percentile of the tree deflection. We fitted a linear model to the relationship between tree deflection and squared wind speed for each tree separately and save the coefficient, the slope of this fit line. We then calculated the ratio of this the slope in summer (defined as 1st May - 31st September) to the slope in winter. For deciduous trees, this ratio indicates whether the increased drag due to leaves in summer causes increased deflection, or whether other mechanisms such as sheltering and damping compensate. We excluded trees with low sample sizes (<10 hours), open-grown trees (due to low sample size) and tropical trees from this analysis. We therefore focussed on the comparison between deciduous forest broadleaves with forest conifers.





As the wind speed increases, both the drag coefficient and damping mechanisms will change (e.g. due to streamlining). The frequency range in which energy transfers from the wind to the tree will therefore shift, and this will be reflected in the slope of the power spectrum. Since our study contains a number of different forest structures as well as open-grown trees, we

cannot assume all sites have similar wind spectra. We therefore study the change in $S_{freq}$ with wind speed for all 103 trees individually. We display these data by fitting a smoothing spline to them.

## 3. Results

### 3.1 Relationship between tree size and fundamental frequency

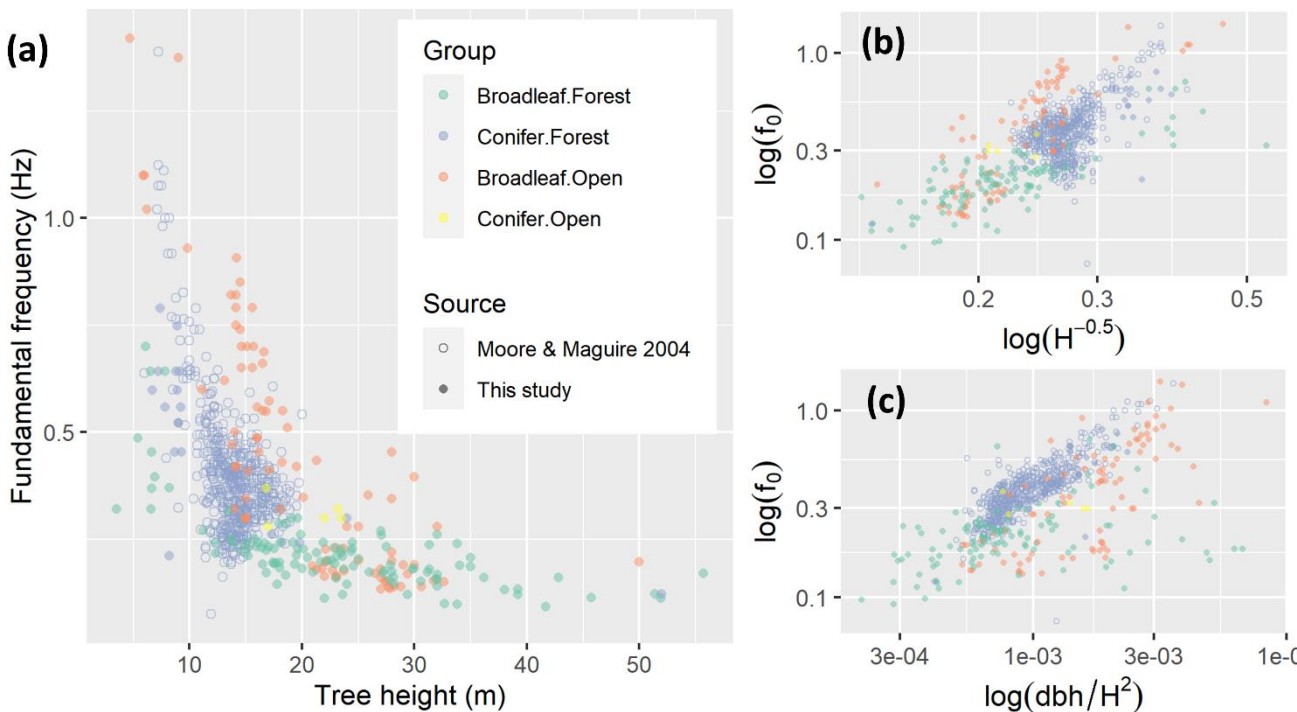


**Figure 2 - a - Fundamental frequency against tree height on a linear scale. Small panels on log-log scales show fundamental frequency against (b) the simple pendulum model and (c) the cantilever beam model.**

Figure 2 shows an overall decrease in $f_0$ with tree height, especially between 5 m and 20 m. These short trees are mostly from conifer forests, the majority of which were included in a previous review (Moore & Maguire, 2004). The open-grown

broadleaves display a similar sharp decrease in $f_0$ with height, although they are offset by approximately 0.5 Hz, which may be due to their wider stems. Table 2 shows that the best predictor of $f_0$ for the conifers was the cantilever beam model $\left( f_0 \sim \right.$



$\frac{dbh}{H^2}$) as demonstrated by (Moore and Maguire, 2004). This is unsurprising, since data from the previous review dominate this group of trees, but this relationship still holds in the new data for forest conifers. However, for both open-grown and forest broadleaves the simple pendulum model $\left(f_0 \sim \sqrt{\frac{1}{H}}\right)$ was the best predictor of $f_0$. It is important to note that the taller trees in

our study are almost exclusively broadleaves, and almost all were from a forest environment. Also, a mixed effects model showed that $f_0$ was the feature most strongly correlated with tree height out of all the features in this study and that tree group, included in the model as a factor, was relatively unimportant (see S2 for details). This confirms that the trends we observed hold across groups as well as within them and do not represent group differences.

| Group | N | Model | $R^2$ | AIC |
|---|---|---|---|---|
| Conifer Forest | 631 | Cantilever beam | 0.67 | -354 |
| | | Simple pendulum | 0.29 | 142 |
| Broadleaf Forest | 111 | Cantilever beam | 0.13 | 78 |
| | | Simple pendulum | 0.61 | -11 |
| Broadleaf Open | 87 | Cantilever beam | 0.55 | 109 |
| | | Simple pendulum | 0.68 | 80 |


**Table 2 - Summary statistics for linear models predicting fundamental frequency using the cantilever beam model and the simple pendulum model for three groups of trees. The data were log-transformed prior to model fitting.**

## 3.2 Differences between types of trees

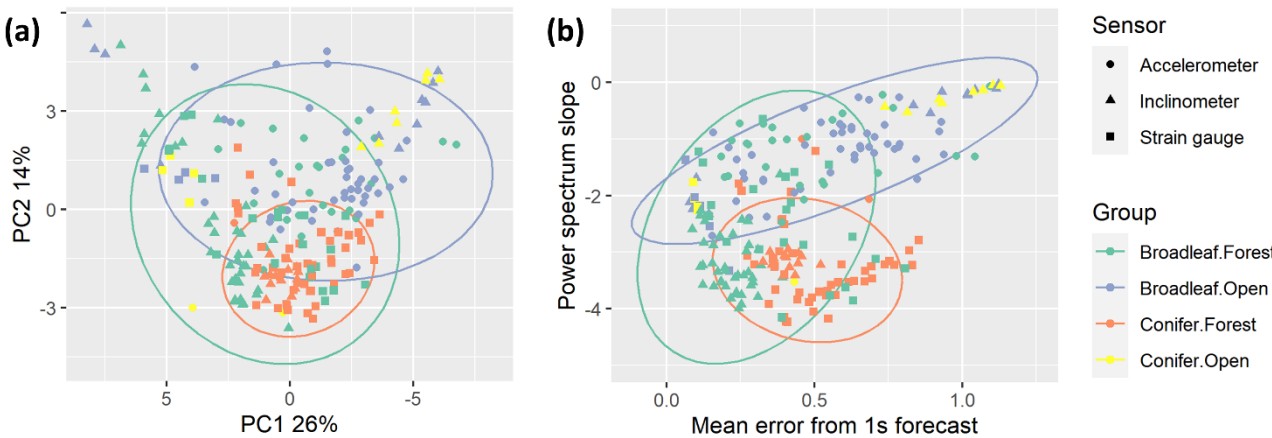


**Figure 3 - Features of tree motion from 225 trees during summer high (local) wind speed conditions projected on (a) first two principal component axes (arbitrary units) and (b) maximally discriminant axes (original units, unscaled). Ellipses represent**




**Student's t-distributions around the data and were only drawn around groups with sufficient samples [broadleaf. forest (n=84), conifer. forest (n=65), broadleaf. open (n=63)].**


Figure 3 shows considerable overlap in the features of all groups of trees in this study. The trees from conifer forests are grouped together and cover a small proportion of the variance in both panels. The open-grown conifers seem to mainly overlap with the open-grown broadleaves, but we did not have enough samples to test this statistically. The forest conifers covered the largest area and overlapped with all other groups. We tested whether sample size affected this pattern by

randomly sampling a subset of the forest broadleaves to the same sample size as the other types and repeating the analysis and found no substantial change in the distribution.

As expected, given that the *catch22* features were designed to be minimally redundant, the principal components did not explain a large amount of the total variation across all features (Figure 3a). The first principal component was driven

by *catch22* features, primarily the mean error from a rolling 3-sample mean forecasting. The second principal component, while it only explained 14% of the variation, helped distinguish between the groups of trees. The second principal component comprised mainly spectral features, specifically $S_{freq}$ and the height of the fundamental frequency peak.

Figure 3b shows how the trees group along the two most discriminant features, determined using linear discriminant analysis

and multinomial regression. It shows a clear separation between forest conifers and open-grown broadleaves, driven by $S_{freq}$, which is related not only to the damping efficiency of the tree, but also to the energy spectrum of the local wind loading (see discussion). We note that a mixed effects model analysis showed that $S_{freq}$ was strongly correlated with *dbh* but that tree group, included in the model as a factor, was not (see S2 for details). The y-axis provides the best separation of forest broadleaves from the other groups based on the mean error from a 1-second forecast (predicting the 4th value from the

previous 3 in a 4 Hz signal). This feature describes how predictable the signal is at short timescales and is likely influenced by both the speed and responsiveness of the tree motion as well as the level of noise in the signal.

We performed multiple classification tasks using a random forest model with evenly balanced groups (N=62), a 50:50 training to testing data split and a 10-fold cross validation approach. In all cases we found that the stream-wise *catch22*

features improved classification ability compared to using only the features from the literature, but that the cross-stream features added minimal further information. We found a moderate classification ability, Cohen's kappa = 0.646, for the three-group classification (omitting open-grown conifers due to lack of data). Two group classification analyses for the forest broadleaves against the other groups had slightly lower accuracy (kappa = 0.533 for open-grown broadleaves and kappa = 0.639 for forest conifers). However, the two-group classification between open-grown broadleaves and forest

conifers resulted in a high accuracy, kappa = 0.90. These results align with our qualitative interpretation of the overlap between groups in Figure 3, as described above.



## 3.3 Changes in tree response with wind speed

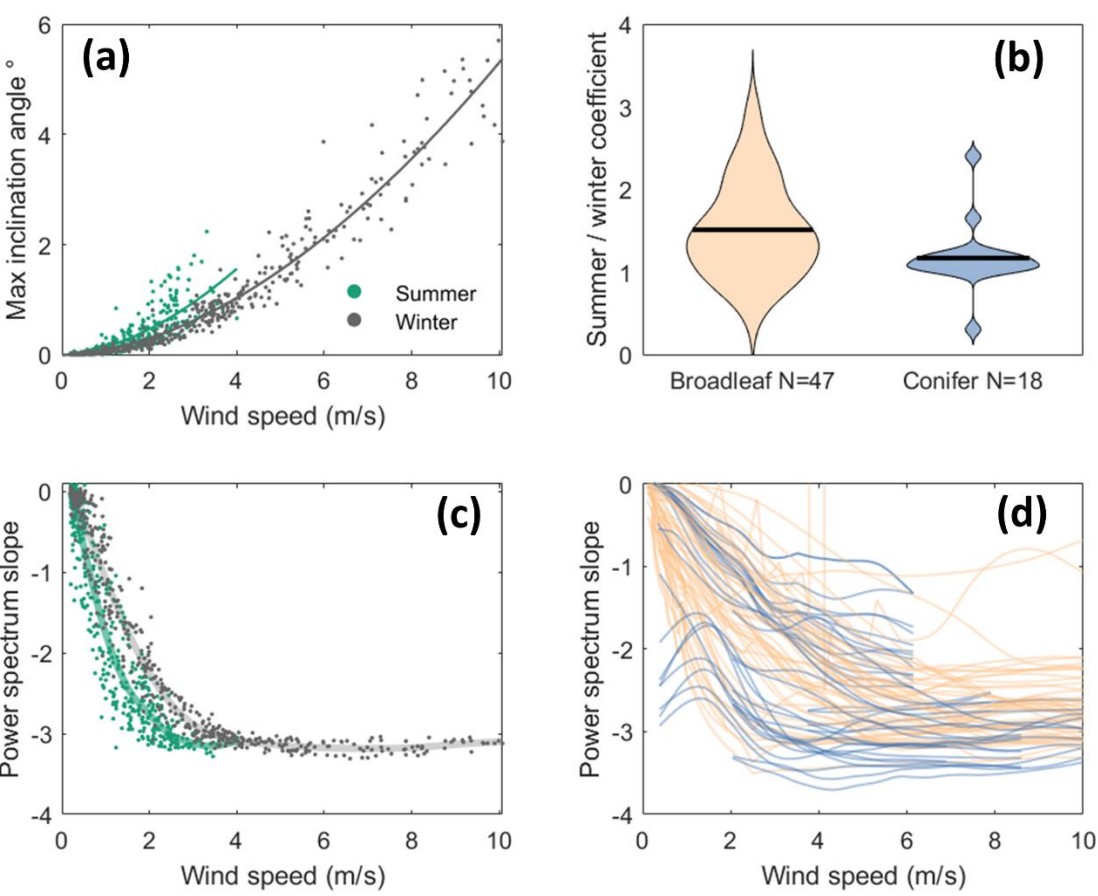

**Figure 4 - Changes in tree motion characteristics with wind speed for trees in forests. a - Tree inclination angle against wind speed**
**for an example deciduous tree (*Acer rubrum* in Orange, Connecticut) in summer and winter, with power law fit lines overlaid. b - Ratio of power law fit slopes in summer and winter for all trees with >10 hours of data in both seasons, excluding tropical trees. The horizontal black lines represent the median value. c - Slope of the power spectrum (between 0.05 and 2 Hz) for a single tree (same as panel a) in summer and winter with smoothing splines overlaid. d - same as panel c but for all trees. Trends are approximated by smoothing splines and summer and winter data are combined.**

Figure 4a shows a typical power-law relationship between tree deflection (in this case measured as inclination) and wind speed. Figure 4b shows how much more deflection occurs for a given wind speed in summer as compared to winter. As expected, we found that the deciduous broadleaves deflection increased more than the conifers (p<0.001) due to the presence of leaves in summer. Interestingly, we also observed an increase in some conifers, possibly due to soil conditions, and a decrease in a few broadleaves, possibly due to sheltering effects. We note that wind speeds were measured outside the forest
or at canopy height in a single location, rather than at the location of each tree in the forest, so that the reported wind speeds represent a local mean rather than the inflow wind speed for each tree.



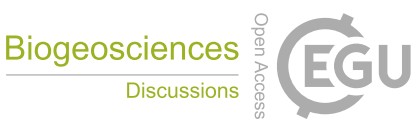

Figure 4c shows a sharp decrease in $S_{freq}$ at low wind speeds, followed by a constant $S_{freq}$ above 3-4 m/s. For this tree we found a slight difference between summer and winter, but there was no systematic difference across all deciduous trees so we combined data from both seasons in the following analysis. Figure 4d shows $S_{freq}$ for all tree with > 10 hours of data (N=93). As in Figure 4c, we see an initial decline followed by a constant $S_{freq}$ above a given threshold. Part of this trend is likely driven by noise at low wind speeds, since the sensors will not reliably measure very low-level tree motion (although the strain gauges were least noisy across the data we analysed). We found no systematic difference in this behaviour between types of trees.

## 4. Discussion

### 4.1 Key features of tree motion

As expected, we found that $f_0$ was a key feature of tree motion, being strongly correlated with tree height (Figure 2 and S3-4) and playing an important role in distinguishing between tree types. We showed that $S_{freq}$ was stable at medium-high wind speeds (Figure 4d) and this was the feature which best distinguished the different types of trees. The fact that two features previously used in the literature remain important when compared to the *catch22* features, which were specifically designed for time-series classification, demonstrates that these two features capture key characteristics of tree motion. Interestingly, the width, height and dominance of the fundamental frequency peak were not important features in this analysis, i.e. they did not explain much of the difference between tree types and were not strongly correlated with tree size.

We also found that a number of *catch22* features were important for classifying tree types and were highly correlated with tree size (S3-4). In particular, the mean error from a rolling 3-sample mean forecast was important in distinguishing tree types (Figure 3b). We interpret this feature as a measure of the regularity of the tree motion at 1 second timescales (all signals were analysed at 4 Hz). This regularity could be related to the wind environment (i.e. a turbulent wind environment leading to lower regularity) or the properties of the tree. For example, a small fast moving conifer would be less predictable and a large broadleaf with complex branching architecture may be less affected by changes wind loading and therefore steadier and more predictable (Rodriguez et al., 2012; Theckes et al., 2011). We found that cross-stream features were superfluous in our analysis, i.e. the features from the literature and stream-wise *catch22* features explained as much of the variability as was possible using this approach.

### 4.2 Differences in motion characteristics between tree types

We were surprised to find no clear separation between open-grown trees and trees in forest in clustering analyses (Figure 3) and a relatively low accuracy in classifying these groups based on their motion. We did, however, observe that the open-grown conifers overlapped mostly with the open-grown broadleaves rather than the forest conifers, but the low sample size

in this group (N=12) hindered any statistical analysis. Given their markedly divergent growth forms (MacFarlane and Kane, 2017), we expected open-grown trees to behave quite differently to the slender trees found in a forest environment (Q2). This lack of difference could be due to the diversity of broadleaves in this sample and difficulty of defining the boundary between open-grown and forest trees (in this study the data owners provided their own interpretation). This negative result may be interpreted as good news from a modelling point of view, since it suggests that separate models are not required for open-grown and forest trees.

We found a consistent and distinct grouping of the forest conifers in both the supervised and unsupervised clustering analyses (Figure 3) and also reflected in the classification analysis. This was driven by their steeper $S_{freq}$, which suggests a difference in the frequency range of the peak wind-tree energy transfer. We also found that the $f_0$ of conifers was best predicted by the cantilever beam model (i.e. a beam with distributed weight) (Moore and Maguire, 2004). Broadleaves, however, were best approximated by the pendulum model (i.e. dominated by the weight of the crown) (Q1). This is likely due to the higher, heavier and more distinct crowns of broadleaves determining the characteristics of their motion more strongly than the distributed crowns of conifers. Together, these findings support the theory that the motion of coniferous trees are relatively similar due to their simple architecture (Jackson et al., 2019), whereas the opposite is true of broadleaves in general. On this basis, we would expect emergent tropical trees, which also have a simple and consistent architecture, to respond to wind in a similarly predictable way. If this is the case (we did not have sufficient data from tall trees to test this theory here) this could enable a simple yet accurate wind damage model for tropical forests focussing only on the large trees, which store the majority of the forest carbon (Bastin et al., 2015). We note that the conifers in this study were considerably smaller than the broadleaf trees. We suggest that the simpler, pendulum model may be valid for a wider range of tree sizes than the cantilever beam model, and it would be interesting to test this for a range of tall conifers.

It is important to note that, in order to maximise the number of trees in our study, the clustering analyses did not incorporate any wind data. Therefore, differences in wind environment may confound differences in tree response. For example, the grouping of the conifers could be explained if patterns of wind flow over conifer forests are generally similar, but wind flow over more broadleaf forests and open-grown trees is more variable. This emphasises the importance of collecting high-resolution (> 1 Hz) wind data alongside tree motion data to understand the details of this interaction. These data are currently lacking for open-grown trees and especially for tropical forests.

### 4.3 Trends with increasing wind speed and seasonal differences

As expected, maximum tree deflection (measured by either acceleration, inclination or bending strain) increased with wind speed (Figure 4a). We found that the gradient of this increase was generally steeper in summer for the deciduous trees (Q3) (Figure 4b), presumably because their leaves provide a larger sail area. This effect is consistent with an increased drag on trees with leaves found by previous studies (Dellwik et al., 2019; Kane and Smiley, 2006). High wind speeds are more

common in winter so losing leaves will reduce a tree's risk of wind damage. However, for some deciduous trees this ratio was below 1, which could be explained by increased sheltering from nearby trees. We also found a slight increase in the tree deflection in summer for some conifers, which could be explained by soil conditions or wood properties, especially in the sites which freeze in winter, although additional measurements would be required to test this.


We observed a remarkably constant $S_{freq}$ from medium to high wind speeds. This plateau in $S_{freq}$ suggests that the frequencies at which energy is transferred from the wind to the tree do not change in the high wind speed regime. This could be because the size of the turbulence structures containing most energy are smaller than the tree crown at high wind speeds, so energy transfer characteristics do not change. This is supported by the difference in $S_{freq}$ between tree types (Figure 3) and the

correlation between $S_{freq}$ and tree size (S4). This plateau in $S_{freq}$ also suggests that no important additional damping and amplification mechanisms emerge above this threshold, meaning that wind damage risk is likely simply to increase with wind speed. We note that the classification analysis and correlation of features with tree size used these stable, high wind speed $S_{freq}$ values.

We could not explain the fact that $S_{freq}$ increased from low to medium wind speeds. We found a wide variation between individual trees in the shape of this curve as well as the stable value, but we found no consistent difference between seasons, suggesting it is not driven by leaf streamlining processes.

### 4.4 Future research directions

A number of research questions arise from this study and the collation of this data set. Most importantly, does wind damage risk increase with tree height? Damage assessments after destructive wind-storms have shown a larger proportion of tall trees uprooted or snapped (Magnabosco Marra et al., 2018; Rifai et al., 2016), which suggests that large trees are more vulnerable to wind damage. However, post-damage assessments may be confounded by other mechanisms, such as disease or lightning, driving the observed pattern in tree mortality. Wind damage risk was measured directly on living trees and found to increase

in two conifer forests (Duperat et al., 2020; Hale et al., 2012) and a tropical forest in Malaysia (Jackson et al., 2020). It would be highly valuable to extend this analysis across all trees monitored in this study to test whether this trend is consistent. However, the different types sensors used to measure tree motion and their mounting positions made this data difficult to interpret in terms of risk across sites. In addition, the low number of trees in each individual experiment limited the applicability of a mixed-effects model in this context. We suggest future studies of this kind combine multiple sensors to

measure tree motion and follow a standardised protocol to calibrate the sensors and so ensure data are comparable between studies. We note that this research question could be answered with local hourly wind speed measurements and therefore does not require sonic anemometers and so could be achieved at low-cost.



Collating this data set highlighted the paucity of data available for open-grown conifers, with the exception of (James et al.,

2006). The low number of these trees in our data set precluded statistical analysis and further data collection in this area would be highly valuable. We also note the low representation of tropical forests, which are likely to be the most complex due to their structure and diversity. Neither of the two tropical sites in this study had high-resolution wind speed data for high wind periods.

A number of single-site studies have demonstrated intriguing relationships between fundamental frequency and ecological processes such as phenology, water status, rain interception and drought (Bunce et al., 2019; Ciruzzi and Loheide, 2019; van Emmerik et al., 2018; Gougherty et al., 2018). Extending these analyses across the full data set would allow us to test how robust these relationships are and therefore whether trees can be used as indicators of ecological processes in other ecosystems. This could be particularly interesting in sites which freeze in winter, since this will have a profound effect on the

wood elasticity. Additional data collection for these purposes should use Inertial Measurement Units (IMUs) which combine accelerometers, inclinometers and magnetometers, because the signals have a lower drift than the strain gauges.

In addition, there has been a long-standing debate on the potential role of resonance between the wind and the tree leading to wind damage (i.e. a high-energy frequency in the wind spectra coinciding with $f_0$) (BLACKBURN et al., 1988; Holbo et al.,

1980; Oliver and Mayhead, 1974; Rodriguez et al., 2008; Schindler et al., 2010; Schindler and Mohr, 2018). We found no evidence for resonance in this study and previous work used singular spectrum analysis to show that the oscillatory component of tree sway diminished with wind speed for four forest Scots Pine trees (Schindler and Mohr, 2018). It would be useful to test whether this result holds across the all the trees studied here.

The subset of data with high-resolution wind speed measurements could be used to explore energy transfer and damping efficiency of different trees. In particular, the identification of $S_{freq}$ as a key feature of tree motion, possibly related to the drag factor and damping efficiency, warrants further exploration (van Emmerik et al., 2018). Further data collection aimed at understanding the mechanisms involved in energy transfer from the wind to the tree should use arrays of sonic anemometers to collect high-resolution (> 5 Hz) local wind speed data. We recommend the use of strain gauges to measure tree motion in

this case, since they provide comparable absolute values which can be related to the risk of mechanical damage.

## 5. Conclusions

In this study, we collated all available data on tree motion in order to explore the key similarities and differences amongst trees as well as possible trends with tree size and wind speed. We compared trees and tested trends based on features

extracted from the tree motion time-series.





We found that trees in conifer forests exhibit similar responses to wind loading, and that these were distinct from those of broadleaves, presumably as a result of their simpler branching architecture. However, we could not accurately distinguish between the motion of open-grown and forest broadleaves, despite the substantial difference in tree shape between the

extremes of this gradient. Our analysis confirms previous studies showing that the fundamental frequency, which describes the speed of tree sway, is a robust feature with which to compare the motion of trees and is strongly related to tree size. Future studies should examine how the fundamental frequency of trees changes over time, since this is related to tree mass and elasticity and hence to a range of important ecological processes such as leaf phenology and water status.

We also found that the slope of the power spectrum, which is related to the wind-tree energy transfer, is an important feature in distinguishing tree types and displayed consistent trends with wind speed. All trees in this study exhibited a remarkably constant slope of the power spectrum from medium to high wind speeds in both summer and winter. This suggests that the relationship between wind loading and tree deflection is simply related to wind speed in the high wind speed range. This result could be an important contribution to estimating wind damage risk and understanding the role of wind in forest

ecology more generally.

**Code availability**

All code created for this analysis is available here: 10.5281/zenodo.4265811. Please contact the corresponding author if you have any questions or need additional materials.

**Data availability**

This data set is a product of the work of many researchers and collaborations are encouraged. All of the data used in this study is available online (currently processing for EIDC). Most of the data is available in a single repository at https://eidc.ac.uk/, which also stores meta-data for all trees used in this study. The Singapore data is available at https://doi.org/10.7910/DVN/FHJBYG, the Montmorency data is available at https://doi.org/10.5683/SP2/WZIKSR, the Manaus data is available at https://doi.org/10.4121/uuid:c9974180-aa9b-40b4-8dbb-06d5b1fce693, the Wytham Woods data

is available at https://doi.org/10.5285/533d87d3-48c1-4c6e-9f2f-fda273ab45bc and the Danum Valley data is available at https://doi.org/10.5285/657f420e-f956-4c33-b7d6-98c7a18aa07a. The Trout Lake and Fond du Lac data are available at http://www.hydroshare.org/resource/38ae9d9fb88d49f9ad2eed1ee07475c0. For any questions regarding data availability please contact the corresponding author.





**Supplement link**

**Competing interests**

The authors declare that they have no conflict of interest

**Author contributions**

TJ, BG and DS conceived the idea. TJ collated the data, which was contributed by JS, AG, DB, DC, SL, MD, TvE, AB, BK, KJ, ED, NA, BG, DS, SK, AW, RS, JM, MR and SvB. TJ conducted the analysis with input from SS, DS, BG, NNA and 470 ED. All authors contributed to the writing.

**Acknowledgments**

TJ and DC are supported by NERC grant NE/S010750/1. SS was support by EPSRC grant EP/R511547/1. TvE was supported by the 4TU Federation project Plantenna. JCR and MD were supported by the Natural Sciences and Engineering Research Council of Canada, grant number RGPIN-2016-05119 and Canadian Wood Fibre Centre, grant CWFC1820-009. 475 SvB was supported by US Dept of Agriculture PR00-NRI-001. SPL and DMC were supported by National Science Foundation grant # EAR-1700983 and US Forest Service agreement #18-JV-11242308-016 under Focus Area 3 of the US EPA Great Lakes Restoration Initiative. MR and VL were supported by NSERC ircsa 233737-98. AG was supported by National Science Foundation Award# IOS-1461868. RS was supported by National Science Foundation, Grants OCE-1600109 and OCE-1853039. BG was supported by National Science Foundation, Grants OCE-1600109 and OCE-1853039.

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
