# Peer review of "The motion of trees in the wind: a data synthesis"

_Biogeosciences, 2020_

## Referee Comment (RC1) · Anonymous Referee #1 · 29 Jan 2021

The paper evaluates the factors influencing the tree responses to wind loading. To that purpose, the Authors resembled an heterogeneous database including tree motion and wind velocity time series over different trees: broadleaf and coniferous in forests and in open environments. Two factors influencing the tree responses emerged: the tree fundamental frequency ($f_0$) and the high-frequency slope of the tree power spectrum ($S_{freq}$). The Authors further found that (1) the fundamental frequency of forest conifers was better predicted according to the cantilever beam model while for broadleaves it was better approximated using a simple pendulum model, and (2) the slope of the tree energy spectrum remained constant from medium to high wind speeds. The paper concludes by some future research directions.

While I find that the Authors did a remarkable job resemble and analyzing this hetero-geneous dataset, I find the results on the tree fundamental frequency not so new as compared to the first Author recent paper (Jackson et al. 2019), and I am quite skepti-

cal about the meaning and importance given in the paper to the slope of the tree power spectrum and about the robustness of its evaluation.

**Major concerns:**

a) In the Introduction section, the Authors give the impression that most of our understanding on the tree fundamental frequency comes from conifers and much less from broadleaves: "Previous data syntheses have focused on the fundamental sway frequency ($f_0$) of conifers [...]. This finding demonstrates that conifers can be approximated by a cantilever beam (i.e. a beam with distributed mass), but it is unclear whether this model extends to other types of trees." However, in a recent study (Jackson et al. 2019) the first author did apparently "a comprehensive view of natural sway frequencies in trees by compiling a dataset of field measurement spanning conifers and broadleaves, tropical and temperate forests" (their abstract). They concluded that "The field data show that a cantilever beam approximation adequately predicts the fundamental frequency of conifers, but not that of broadleaf trees" (still their abstract). I am, therefore, wondering what is new in this paper compared to Jackson et al. (2019) on the tree fundamental frequency? Does it require a new publication? What are the differences between the dataset used in both papers? The Authors should better position their paper compared to the previous one.

b) I find the meaning of the slope of the tree energy spectrum not clear in the paper. In lines 94-95, it is written that "the slope of the power spectrum ($S_{freq}$) can be used as an overall measure of energy transfer between wind and tree at different frequency ranges (van Emmerik et al., 2018; Van Emmerik et al., 2017)". I am not sure to agree with this statement that $S_{freq}$ represents the energy transfer between wind and tree. In my opinion, it is more representative of the tree energy transfer (cascading) or damping from $f_0$ to high frequencies. Indeed, $f_0$ is usually located at the level of the inertial subrange of the wind velocity spectrum (see Figures S6 and S7), i.e. at frequencies larger than the frequencies of the main eddy motions at canopy top. I would think that the energy transfer between wind and tree occurs mainly at lower frequencies than $f_0$,

where the tree power spectra exhibit the same distribution with frequency as the wind spectra. I think $S_{freq}$ reflects how the tree damps/transfers its energy independently of the wind. Maybe a way to verify which flow motions are involved in tree motions is to look at the momentum flux cospectrum, assuming that the momentum flux at canopy top is totally absorbed by the trees. For example, if you look at Figs 4 and 6 of Dupont et al. (2018, Agric Forest Meteorol., 262, 42-58), you can see that most of the canopy-top momentum flux occurs at frequencies lower than $f_0$. Smaller eddies than the dominant canopy-top eddies may transfer as well energy to the tree but I would think it mainly concerns branches and less the trunks where the measures presented in this paper have be done. Branch motions are not necessarily in phase with the trunk motions. The lower $S_{freq}$ for broadleaves than for conifers may just reflect their difference in architecture. I would think that $S_{freq}$ is representative of the tree properties, but not representative of the wind. Is it really new/surprising to observe differences between tree species in energy cascading/damping knowing that this mechanism depends on the tree properties (architecture, stiffness...)?

c) I am skeptical about the estimation of $S_{freq}$. First, tree energy spectra do not show scaling law between the fundamental frequency and high frequencies because of the presence of secondary maxima. It is therefore quite questionable to define a slope there. Second, this slope has been defined for a specific frequency range in Hz (lines 167-168), while this frequency range should start from a frequency depending on $f_0$. The estimated slope is certainly sensitive to the height and width of the tree spectrum fundamental and secondary maxima. There are many cases where it seems impossible to define $S_{freq}$ (see the tree spectra in Figures S6 and S7). I am, therefore, not surprise to see some erratic behaviors in $S_{freq}$ in Figures 4d. At least, these erratic behaviors should have been removed. Third, the Authors seem surprised and present as a result the fact that below a threshold wind speed value, $S_{freq}$ decreases with wind speed (Figures 4c-d). In my opinion, this decrease of $S_{freq}$ reflects the increasing noise of the tree data at high frequencies as the wind diminishes. With decreasing wind, the frequency of the main canopy motions gets lower. Consequently, $f_0$ is shifted to the

bottom (high frequencies) of the inertial subrange of the wind velocity spectrum, where there is much less energy. The high-frequency trunk motions become negligible. I am, therefore, not surprise to see that $S_{freq}$ decreases with wind speed, its evaluation becomes irrelevant and should not be presented.

d) The location of the wind speed measurement should be clarified. The Authors compared the tree inclination angle against the wind speed between summer and winter but they do no say clearly where the wind has been measured (at canopy top, outside the plot. . .). It is just written that "We note that wind speeds were measured outside the forest or at canopy height in a single location" (Lines 304-305). This difference in wind speed measurements between experiments makes it difficult any comparison. For measurements outside the forest, are winter and summer measurements representative of the same footprint? It is difficult to conclude on Fig 4a because we do not know where the wind speed has been measured. The wind speed should be normalized by a reference wind speed.

**Specific comments:**

1) Line 87: Which balance are you talking about? Can you be more specific?

2) Lines 97 and 46: "This study brings together all available data on tree motion". This is quite a strong statement. I see at least two datasets that were not considered or mentioned in this study: Sellier et al. (2008, Forestry 81, 279–297) and more recently Dupont et al. (2018, Agric Forest Meteorol., 262, 42-58).

3) Line 102: Correct the parentheses for the Lubba et al. reference.

4) Line 124: "We therefore focus on analyses which do not require these data (although we explore this data in supplementary S2)." This is confusing, why talking about these data if you did not use them?

5) Lines 138-139: "although open-grown trees exposed to strong winds may experience slowly varying displacements due to the mean wind speed on this timescale

(Angelou et al., 2019; James et al., 2006)." I do not understand. Could you clarify?

6) Line 144: How did you define the windy conditions?

7) Lines 227-229: "The frequency range in which energy transfers from the wind to the tree will therefore shift, and this will be reflected in the slope of the power spectrum." I do not understand why it will be reflected in the slope of the power spectrum. Shift in which direction?

8) Line 276: "It shows a clear separation between forest conifers and open-grown broadleaves, driven by $S_{freq}$, which is related not only to the damping efficiency of the tree, but also to the energy spectrum of the local wind loading". I do not understand the justification for the last part of this sentence.

9) Lines 328-329: "This regularity could be related to the wind environment (i.e. a turbulent wind environment leading to lower regularity)". I do not understand the notion of "wind environment", nor what is written between parentheses.

10) Line 342: "separate models", to which models are you referring to?

11) Line 347: "which suggests a difference in the frequency range of the peak wind-tree energy transfer." Could you clarify? I do not understand.

12) Line 378: "This could be because the size of the turbulence structures containing most energy are smaller than the tree crown at high wind speeds". I do not think so. The main turbulent motions at canopy top should not change much size with wind speed. In my opinion, the plateau of $S_{freq}$ just shows that $S_{freq}$ does not inform on the wind-tree energy transfer but only or mainly on the energy cascading/dissipation of the tree motions from $f_0$ to high frequencies, which only depends on the tree properties and much less on the wind intensity.

13) Line 422: "the oscillatory component of tree sway diminished with wind speed for four forest Scots Pine trees". I am not sure to understand this sentence and how it demonstrates the presence of a resonance mechanism.

14) Lines 423: "the all the trees", please rephrase.

15) Line 447-448: "All trees in this study exhibited a remarkably constant slope of the power spectrum from medium to high wind speeds in both summer and winter. This suggests that the relationship between wind loading and tree deflection is simply related to wind speed in the high wind speed range." So, it does not depend on the tree properties? I would say that it shows that $S_{freq}$ depends on tree trunk and branches properties and less on the presence or not of leaves.

16) Figures S6-S7: Is it the same graduations for the vertical axis of the tree and wind energy spectra? Could you show the graduations? It would be nice to show the -2/3 slope. Some velocity energy spectra look very flat but it may just be a question of graduation.

———————————————

---

## Short Comment (SC1) · 29 Jan 2021

I have been asked to post a short comment after declining to review. This is just my view on the manuscript as a scientist with interests in the research topic. I do not express any opinion on acceptance for publication or not.

The study presents the analysis of tree motion over a large and heterogeneous dataset. The results identify broad response patterns in relation to trees' architectural archetype. This preliminary classification should be useful in forest ecology and this type of meta-analysis is important for global studies of wind risk. In contrast, the number of tree samples in the study is very low compared to the combination of shape, size and species that is possible within each archetype.

The analysis introduces generic time series classifiers (catch22) in addition to more conventional descriptors of tree motion series (such as $f_0$). I find this addition novel and

useful because it does not depend on *a priori* knowledge of tree motion. The approach has the potential to tease out new features of tree dynamics. The use of machine learning for predicting tree size based on motion characteristics is also interesting. It is an important step towards integrating vibrational data in plant phenotyping.

Three points worth noting:

- I understand the need to harmonize the datasets, but I do not think that degrading the temporal resolution of high quality signals was necessary. A 4-Hz sampling may be sufficient to study the movement of a slow-moving tree but detrimental in the case of a fast-moving one. Down sampling all data to 4 Hz may have introduced artifacts and caused loss of information.

- The paper erroneously states twice that the study contains all tree motion data. There are other tree motion datasets that have not been included in this study.

- L348. Poor phrasing. A beam is a system with distributed properties. A cantilever beam is, typically, a beam supported at one end and free at the other.

---

## Short Comment (SC2) · 29 Jan 2021

Thanks for your input and your favourable comments about our research.

I agree that, while we have collected together lots of data, we have not covered the full range of tree morphology. The largest data gaps are for tropical forests, we currently have only 19 trees in Malaysia and 19 trees in Brazil. We are also missing data for open-grown conifers and conifers above 30 m tall. More data for these groups would certainly improve our analysis and we encourage future studies to focus on these ecosystems (I also intend to collect more tropical forest data myself).

On a related subject, you stated that we do not include all tree motion data. If you know of any other data sets not included here, please let us know. A secondary purpose of this paper is to describe and collate all available tree motion data sets and deposit them online for future studies. I am currently working on the data repository which

will be published alongside this paper. I would note that we are aware of one other data set which is private and therefore couldn't be included, and the text therefore says 'all available data'. Nevertheless, we would be happy to change this phrasing in the manuscript.

Thanks for your comments on down-sampling all time series to 4Hz. However, I believe this was a necessary precaution for this type of analysis. Many of the time series features are sensitive to the sampling frequency, and therefore may distinguish between tree motion time series based on this, rather than the characteristics of the tree motion. Down-sampling inevitably results in loss of information, but I do not think this is too severe. Figure 2a shows that only four of the trees used in our analysis have a fundamental frequency above 1 Hz and the majority of trees have a fundamental frequency under 0.5 Hz. This means that all but four trees have >4 samples per oscillation, while most trees have >8 samples per oscillation. We believe this is sufficient for the current analysis, and preferable to including different sampling frequencies. I would note that the data will be deposited online in the original sampling frequencies so that future studies can decide whether or not to harmonize in this way.

Thanks for your correction on the definition of a cantilever beam, I will correct this.

---

## Short Comment (SC3) · 4 Feb 2021

Thank you for your review. I am discussing your points with co-authors and we will make a joint reply in due course. For the time being, I am replying personally as first author.

I think that some of the issues you raise, while appropriate for a simple single site study, are necessary complications when conducting a multi-site study such as this. For instance, some studies did not record wind data at all, others used a MET station near the forest to give hourly mean wind speeds and a few studies collected wind data at canopy height at 10 Hz using a 3D sonic anemometer. These differences preclude some types of analysis that would otherwise have been valuable. However, I think it is still useful to bring together these diverse data sets and compare them. Many of our conclusions (see 'Future work directions') address these differences between sites

and we suggest best practice for future studies.

The aims of this paper were (1) to bring together numerous tree motion data sets; (2) to test whether previous results hold across multiple sites and (3) to describe the key similarities and differences that emerge when studying tree motion at this scale. I will respond to your points in order below.

Major concerns:

a) As you point out, there is certainly some overlap between this paper and Jackson et al (2019). I agree with you that this analysis should be better positioned with respect to Jackson et al (2019) and the key differences explicitly stated. However, Jackson et al (2019) was primarily a simulation study, using finite element analysis to explore the relationship between tree architecture and natural sway frequencies, and it used the field data only to demonstrate the range of natural variation. Furthermore, there are three key differences between the analysis in Jackson et al (2019) and the current study:

(1) The data set we presented in Jackson et al (2019) was comprised of only summary data on the fundamental sway frequency, calculated using different methodologies in each individual study. In the current study, we have collated the raw tree motion data and calculated the fundamental sway frequency using the same method (wavelet analysis) for each time-series. This makes the current analysis more robust.

(2) There is more data in the current study. Comparing table S1 from Jackson et al 2019 with Table 1 of the current study shows 39 additional trees from conifer forests, 8 additional open-grown conifers, 61 additional open-grown broadleaves and 17 additional trees from broadleaf forests. The 61 additional open-grown broadleaves, which were under-represented in Jackson et al (2019) make a substantial contribution to the current study.

(3) We did not test the simple pendulum model in Jackson et al (2019), which turns out

to be quite important in the current study.

I believe that, given the aims of this paper, it is valuable to include the fundamental frequency analysis (Figure 1) despite the overlap with Jackson et al (2019). This analysis gives the reader a good understanding of the variation in tree motion across our data set (i.e. the 'slowest' tree in our study takes approximately 10 seconds to complete one oscillation, while the 'fastest' takes approximately 0.7 seconds). It is also an important feature of tree motion that has been widely used in the literature and would therefore be a strange omission from a paper which aims to synthesis tree motion data. I will adapt the text to reflect these updates.

b) I agree with you that the momentum flux cospectrum, or the transfer function would be preferable, but this was not possible for the majority of trees in this study due to the lack of high-resolution wind data. Most studies of tree motion do not include high-resolution wind data (L121-125) because of the prohibitive costs, we discuss this in the 'Future research directions' section of the discussion. We included the slope of the power spectrum in this study because that two previous studies (van Emmerik et al 2017 and 2018) found it to be important and useful in the absence of high-resolution wind data. Given our results, that it was an important factor in distinguishing types of trees, I think its inclusion was justified.

The interpretation of the slope of the power spectrum is something we discussed at length while writing the manuscript. I will therefore respond to your comment at greater length after discussing it with my co-authors. However, in response to your query, I would point out that this result is certainly new (to the best of our knowledge) and shouldn't have to be 'surprising'.

c) I explored different methods to estimate the slope of the power spectrum in this study and found that using a large fixed interval was the most robust. This is similar to the approach taken in previous publications (van Emmerik et al 2017 and 2018). As you suggest, defining a different frequency range for each tree based on its fundamental

frequency is an attractive idea, but in practice this method was too noisy to be applied across such a diverse range of data. A number of trees in our study exhibited no consistent fundamental frequency peak, so the automatically extracted slope of the power spectrum would be undefined in these cases. However, I am happy to remove the most erratic lines from figure 4d.

The definition we use gives a measure of the decline in energy in the tree spectra from 0.05 to 2 Hz. We find a rather consistent pattern across all trees and our results show this to be an important feature.

I will address your third point (whether the decreasing trend is driven by noise) after discussion with the co-authors. We considered this possibility while writing the manuscript and decided otherwise.

d) The location of the wind speed measurement is different for each study (L121-125), this is one of the challenges in working with such diverse data sets. Importantly, Figure 4d shows the changes over time for each tree individually, we are not comparing the magnitude of wind speeds across sites. It is these changes with increasing wind speed that we compare and find to be remarkably similar.

Part of the value of this study is that, in bringing together these data sets, we can compare the advantages and disadvantages of different experimental setups. I can include more information on the location of the wind sensor for each site individually in table 1. Furthermore, the online data repository will include descriptions of the individual sites.

RE figure 4a – in this case a single tree is presented as an example and the wind speed measurement was taken outside the forest at a nearby MET station. I will include this information in the figure caption. Thank you for pointing this out.

I don't understand what you mean by 'the wind speed should be normalized by a reference wind speed'. In each site wind speed was measured in a different way, therefore a 5 m/s wind speed measured outside the forest in site A is not the same as a 5 m/s

measured at canopy top in site B. It would be nice to work out how these two measurements compare, perhaps with reference to some standard measure of wind speed 100 m above the surface. However, I do not think methods exist to make this conversion across sites from cities to tropical forests. If you have suggestions / corrections on this I would be happy to learn more.

Specific comments:

In this section I will again leave comments which mainly concern the slope of the power spectrum until I have discussed them with co-authors.

1) Thank you for pointing this out. This is the balance between energy transfer from the wind to the tree, and energy dissipation by damping. I will update the text to explain this explicitly

2) Thanks, the previous comment from Damien Sellier also mentioned this. I will change this statement in the text to: 'a large data set of available tree motion data' or similar. Additionally, I will reach out to these authors and attempt to include their data in the study. As I understand it, these studies represent 2 and 3 medium sized conifers, respectively. This is the group of trees best represented in our sample so their inclusion is highly unlikely to change the results. I could also include these data in the online repository for future use. Also, we are aware of three other small data sets, but they were not available to this study.

3) I will correct this.

4) This refers to the fact that different data sets have different wind data associated with them (L121-125). This is a key point in understanding the challenges of this type of multi-site study, as discussed in detail above. This confusion may have contributed to some of the 'Major Concerns' discussed above. I mention these high-resolution wind data because they are the 'gold standard' in experimental design and many of the more recent papers on this subject rely on these data. I suggest that, instead of omitting this

important point, I expand upon it and lay out what the lack of these data means for the subsequent analysis.

5) When processing raw tree motion data, it is generally necessary to define a zero or mid-point at which the tree is vertical. This allows the data to be interpreted in terms of displacement from this position in different directions. This is complicated by the fact that some motion sensors 'drift' over time, i.e. an offset builds up due to a number of factors (L133-134). Most previous studies have done this by assuming that the tree will pass through its mid-point often. Therefore, we subtract a running mode or apply a high-pass filter to each interval of data (1 hour or 10 minutes are commonly used) to correct for this offset. This has been shown to be effective in a number of studies and is standard practice for trees in forests. However, open-grown trees may behave differently. In particular, they may be displaced from the vertical for a long period of time due to the effect of the mean wind speed. This is discussed in detail in the cited paper (Angelou et al 2019) and we mention it here as a caveat to our results. I will explain this in greater detail in the text, thanks for pointing it out.

6) I chose the windiest available period for each data set. I will update the text to reflect this.

7) I will address this after discussing with co-authors.

8) I will address this after discussing with co-authors.

9) Perhaps this is a poor choice of phrasing on my part. By 'wind environment' I am referring to the wind conditions affecting a tree or group of trees. For example, tree A, situated in a dense forest will experience turbulent wind conditions with most of the loading on the upper canopy, while the lower parts of the tree are sheltered. Tree B, growing on a flat coastline with no other trees or obstacles nearby will experience consistent wind speeds and the wind loading will be distributed across the height of the tree. In the parentheses I am suggesting that the motion of tree A may be more regular than that of tree B.

10) Sorry for the confusion, I will elaborate and make this point clear in the text. There have been a number of attempts to model tree response to wind loading (e.g. Forest-Gales) but these are mostly based on uniform stands of conifer plantations. It would be valuable if these models could be transferred to more 'natural' forest environments and to open-grown trees. If we had found a clear separation between different types of trees in Figure 3, i.e. they move in distinct ways, transferring these models between these types of trees would have been unlikely to work. We therefore suggest it is good news, from a modelling perspective, that the trees overlap in Figure 3.

11) I will address this after discussing with co-authors.

12) I will address this after discussing with co-authors.

13) Thanks for pointing this out – I will expand upon it in the text. This sentence is describing the work in the cited paper (Schindler and Mohr, 2018) and suggests that no resonant effects were found. They used singular spectrum analysis to separate the oscillatory components of tree sway and found that their importance diminished wind increasing wind speed. We are suggesting that this analysis should be carried out across our newly collated data set, in order to test whether this result holds more generally.

14) Sorry about that, I will correct it.

15) I will address this after discussing with co-authors.

16) I am happy to adjust the figures in the supplementary materials – thanks for your suggestion. Comments outstanding

In summary, I have addressed a number of your comments above but have left those related to the slope of the power spectrum for discussion with co-authors. To help keep track of this, I have pasted your comments which are still outstanding below. Thank you again for your review of our paper, and I will respond to the following comments in the near future.

Major concerns:

b) I find the meaning of the slope of the tree energy spectrum not clear in the paper. In lines 94-95, it is written that "the slope of the power spectrum (Sfreq) can be used as an overall measure of energy transfer between wind and tree at different frequency ranges (van Emmerik et al., 2018; Van Emmerik et al., 2017)". I am not sure to agree with this statement that Sfreq represents the energy transfer between wind and tree. In my opinion, it is more representative of the tree energy transfer (cascading) or damping from f0 to high frequencies. Indeed, f0 is usually located at the level of the inertial subrange of the wind velocity spectrum (see Figures S6 and S7), i.e. at frequencies larger than the frequencies of the main eddy motions at canopy top. I would think that the energy transfer between wind and tree occurs mainly at lower frequencies than f0, where the tree power spectra exhibit the same distribution with frequency as the wind spectra. I think Sfreq reflects how the tree damps/transfers its energy independently of the wind. Maybe a way to verify which flow motions are involved in tree motions is to look at the momentum flux cospectrum, assuming that the momentum flux at canopy top is totally absorbed by the trees. For example, if you look at Figs 4 and 6 of Dupont et al. (2018, Agric Forest Meteorol., 262, 42-58), you can see that most of the canopy-top momentum flux occurs at frequencies lower than f0. Smaller eddies than the dominant canopy-top eddies may transfer as well energy to the tree but I would think it mainly concerns branches and less the trunks where the measures presented in this paper have be done. Branch motions are not necessarily in phase with the trunk motions. The lower Sfreq for broadleaves than for conifers may just reflect their difference in architecture. I would think that Sfreq is representative of the tree properties, but not representative of the wind. Is it really new/surprising to observe differences between tree species in energy cascading/damping knowing that this mechanism depends on the tree properties (architecture, stiffness: : :)?

Part of c) Third, the Authors seem surprised and present as a result the fact that below a threshold wind speed value, Sfreq decreases with wind speed (Figures 4c-d). In my

opinion, this decrease of Sfreq reflects the increasing noise of the tree data at high frequencies as the wind diminishes. With decreasing wind, the frequency of the main canopy motions gets lower. Consequently, f0 is shifted to the bottom (high frequencies) of the inertial subrange of the wind velocity spectrum, where there is much less energy. The high-frequency trunk motions become negligible. I am, therefore, not surprise to see that Sfreq decreases with wind speed, its evaluation becomes irrelevant and should not be presented.

Specific comments 7) Lines 227-229: "The frequency range in which energy transfers from the wind to the tree will therefore shift, and this will be reflected in the slope of the power spectrum." I do not understand why it will be reflected in the slope of the power spectrum. Shift in which direction?

8) Line 276: "It shows a clear separation between forest conifers and open-grown broadleaves, driven by Sfreq, which is related not only to the damping efficiency of the tree, but also to the energy spectrum of the local wind loading". I do not understand the justification for the last part of this sentence.

11) Line 347: "which suggests a difference in the frequency range of the peak wind-tree energy transfer." Could you clarify? I do not understand.

12) Line 378: "This could be because the size of the turbulence structures containing most energy are smaller than the tree crown at high wind speeds". I do not think so. The main turbulent motions at canopy top should not change much size with wind speed. In my opinion, the plateau of Sfreq just shows that Sfreq does not inform on the wind-tree energy transfer but only or mainly on the energy cascading/dissipation of the tree motions from f0 to high frequencies, which only depends on the tree properties and much less on the wind intensity.

15) Line 447-448: "All trees in this study exhibited a remarkably constant slope of the power spectrum from medium to high wind speeds in both summer and winter. This suggests that the relationship between wind loading and tree deflection is simply

related to wind speed in the high wind speed range." So, it does not depend on the tree properties? I would say that it shows that Sfreq depends on tree trunk and branches properties and less on the presence or not of leaves.

---

## Author Comment (AC1) · 26 Feb 2021

Dear reviewer, thank you again for taking the time to comment on our manuscript. I have discussed your comments with co-authors and respond to them below. I believe most of the smaller issues were dealt with in my previous reply, so this response deals only with issues arising from the interpretation of the slope of the tree motion power spectrum (Sfreq).

I have broken down your major concern b) into three parts, which I will address in turn:

Reviewer comment b.1)

I find the meaning of the slope of the tree energy spectrum not clear in the paper. In lines 94-95, it is written that "the slope of the power spectrum (Sfreq) can be used as an overall measure of energy transfer between wind and tree at different frequency

ranges (van Emmerik et al., 2018; Van Emmerik et al., 2017)". I am not sure to agree with this statement that Sfreq represents the energy transfer between wind and tree. In my opinion, it is more representative of the tree energy transfer (cascading) or damping from f0 to high frequencies.

Author response b.1)

Thanks for raising this issue. We agree with you that Sfreq will be influenced by the damping and related to the stiffness of the tree. However, unlike typical building structures, the stiffness and damping of the system will change with wind speed (particularly the aerodynamic damping), as the tree changes shape and the whole system deforms.

The spectral response of the tree is essentially the spectra of the wind modified by the tree response (close to a lumped mass damped harmonic oscillator for conifers). This is described in equation 27 in Gardiner 1992. This is also presented in Mayer (1987), Kerzenmacher and Gardiner (1998) and Sellier et al (2008). It seems that many of your criticisms stem from this difference in interpretation and it would be useful if you could provide some references to support your view.

We appreciate that the current wording may be unclear we suggest revising this to: 'the tree spectrum is essentially the wind spectrum modified by the tree response. Therefore, the slope of the tree spectrum (Sfreq) is the result of the energy transfer between wind and tree as well as the energy transfer within the tree itself'.

Reviewer comment b.2)

Indeed, f0 is usually located at the level of the inertial subrange of the wind velocity spectrum (see Figures S6 and S7), i.e. at frequencies larger than the frequencies of the main eddy motions at canopy top. I would think that the energy transfer between wind and tree occurs mainly at lower frequencies than f0, where the tree power spectra exhibit the same distribution with frequency as the wind spectra. I think Sfreq reflects how the tree damps/transfers its energy independently of the wind. Maybe a way to

verify which flow motions are involved in tree motions is to look at the momentum flux cospectrum, assuming that the momentum flux at canopy top is totally absorbed by the trees. For example, if you look at Figs 4 and 6 of Dupont et al. (2018, Agric Forest Meteorol., 262, 42-58), you can see that most of the canopy-top momentum flux occurs at frequencies lower than f0. Smaller eddies than the dominant canopy-top eddies may transfer as well energy to the tree but I would think it mainly concerns branches and less the trunks where the measures presented in this paper have be done. Branch motions are not necessarily in phase with the trunk motions.

Author response b.2)

The aim of our manuscript is not to identify which frequency ranges are most important, but to study the similarities / differences between trees. This helps contextualize the more detailed, single site studies. As explained in my previous response, the fact that this study contains a number of diverse data sets (which is its strength) precludes the analysis you suggest, which would require high resolution wind data for all sites.

All frequencies in the wind spectra necessarily stimulate tree motion, albeit rather unequally (this response is called the mechanical transfer function or admittance function). Previous studies have analysed which frequencies contribute most to this energy transfer (e.g. Dupont et al. 2018 and Gardiner 1995, Schindler 2008, Schindler and Mohr 2019). It is true that the dominant motion is triggered by the coherent turbulent structures in the wind (Schindler and Moher, 2019) but this does not mean there is no response at other frequencies. The wind drag will be primarily on the leaves / needles, resulting in their motion which will transfer to the stem. Motion also passes from the trunk back to the branches and to the leaves and is then dissipated as vortex shedding from the leaf tips (Spatz and Theckes, 2013). This leads to a spectral short-cut in the wind spectra inside the canopy (Finnigan, 2000). Therefore, although the spectral shape of the tree displacement does reflect this energy transfer, it is mixed with the direct response to the wind at all frequencies. We observe, therefore, a superposition of tree fundamental mechanical response and a supplementary response due to the

transfer of energy between different frequencies that overall leads to a transfer from low frequency motion to high frequency needle / leaf waving.

Reviewer comment b.3)

The lower Sfreq for broadleaves than for conifers may just reflect their difference in architecture. I would think that Sfreq is representative of the tree properties, but not representative of the wind. Is it really new/surprising to observe differences between tree species in energy cascading/damping knowing that this mechanism depends on the tree properties (architecture, stiffness: : :)?

Author response b.3)

We agree with you that some (or even most) of the differences between trees arise from their different architectures. To our knowledge, no study has compared tree motion spectra across multiple species and different genera and growing conditions before. Our aim was not to produce surprising results, but to test whether or not different trees behaved in different ways. Perhaps the surprising part (and not what we expected initially) is the degree of similarity across such a diverse data set. Sfreq is rather consistent across all trees and decreases to a value of -3 at around 4 m/s in most cases. We do not speculate in the paper what this indicates but it might suggest a convergent evolutionary response to the danger of wind loading and an efficient method for dissipating energy.

Reviewer comment c)

Third, the Authors seem surprised and present as a result the fact that below a threshold wind speed value, Sfreq decreases with wind speed (Figures 4c-d). In my opinion, this decrease of Sfreq reflects the increasing noise of the tree data at high frequencies as the wind diminishes. With decreasing wind, the frequency of the main canopy motions gets lower. Consequently, f0 is shifted to the bottom (high frequencies) of the inertial subrange of the wind velocity spectrum, where there is much less energy. The

high-frequency trunk motions become negligible. I am, therefore, not surprise to see that Sfreq decreases with wind speed, its evaluation becomes irrelevant and should not be presented.

Author response c)

We do not have a good explanation for this change and do not present it as a key result (L385). However, we do not think that this is simply due to a declining signal-to-noise ratio. Even at relatively low wind speeds, many trees have large motions orders of magnitude larger than the sensitivity of the sensors. Obviously, our study contains a wide range of sensors and some of them may be noisy, but it also contains some extremely high-quality data sets which exhibit the same pattern. For example, accelerometers are extremely sensitive and the strain gauges are able to monitor the tiny diurnal fluctuations (few millimetres) in stem swelling as the trees stop respiring at night (Duperat et al, 2020).

Specific comments.

I will not copy and paste each comment below, instead I refer to them by number. I only respond to those not previously addressed.

7) The tree spectrum is essentially the wind spectrum modified by the tree response. Sfreq is the slope of the tree spectrum. If either the wind spectrum or the tree response changes (due to increased wind speed or streamlining, respectively) we expect Sfreq to change. As for the direction, an increasing wind speed should lead to more energy at higher frequencies.

8) As discussed above, the tree spectrum is essentially the wind spectrum modified by the tree response. We state explicitly that Sfreq will depend on the wind spectrum because different sites may have different spectra (i.e. a uniform conifer forests compared to a multi-layered tropical forests or an open-grown tree in a park).

11) This is the same issue as comment (8). I will clarify this at the revision stage.

12) We presented this mechanism as a possible explanation for our observations, it is purely speculative. We are happy to delete this speculation from the discussion.

15) The value of Sfreq does depend on tree properties, we can see this in our comparison across trees (Fig 2b). We are arguing that the lack of changes in Sfreq suggests there are no substantial changes in the tree response, such as additional damping mechanisms or resonant effects.

Duperat, M., Gardiner, B., Ruel, J.-C., 2020. Testing an individual tree wind damage risk model in a naturally regenerated balsam fir stand: potential impact of thinning on the level of. For. An Int. J. For. Res. 1–10. https://doi.org/10.1093/forestry/cpaa023

Dupont, Sylvain, et al. "How stand tree motion impacts wind dynamics during windstorms." Agricultural and Forest Meteorology 262 (2018): 42-58.

Gardiner BA (1992) Mathematical modelling of the static and dynamic characteristics of plantation trees. In: Franke J, Roeder A (eds) Mathematical modelling of forest ecosystems. Sauerländer, Frankfurt/Main, pp 40–61

Gardiner, B. A. "The interactions of wind and tree movement in forest canopies." Wind and trees (1995): 41-59.

Kerzenmacher, Tobias, and Barry Gardiner. "A mathematical model to describe the dynamic response of a spruce tree to the wind." Trees 12.6 (1998): 385-394.

Mayer H (1987) Wind induced tree sways. Trees 1: 195–206

Schindler, D., 2008. Responses of Scots pine trees to dynamic wind loading. Agric. For. Meteorol. 148, 1733–1742. https://doi.org/10.1016/j.agrformet.2008.06.003

Schindler, D., Mohr, M., 2019. No resonant response of Scots pine trees to wind excitation. Agric. For. Meteorol. 265, 227–244. https://doi.org/10.1016/j.agrformet.2018.11.021

Sellier, Damien, Yves Brunet, and Thierry Fourcaud. "A numerical model of tree aerodynamic response to a turbulent airflow." Forestry 81.3 (2008): 279-297.

---

## Referee Comment (RC2) · Anonymous Referee #2 · 26 Mar 2021

In the study, the authors present an analysis of tree motion based on a large sample basis that they collected from several sources (including both deciduous and coniferous trees). They found that the most important features for describing tree motion are the fundamental frequency (f0) of a tree and the slope of the power spectrum (Sfreq), and that Sfreq is constant at medium-high wind speeds for all trees included in the study. This means that wind damage risk is simply related to wind speed. Moreover, they found that f0 is strongly related to tree architecture.

In general, I found the study clearly written and that it presents a valuable contribution to existing research and motivation for further studies. I would like to note that I have read the already existing reviewer comments and the authors' responses. I had similar concerns about some of the issues mentioned in one review (e.g., large, heterogeneous database; increasing noise of the tree data at low wind speeds). However, I

appreciate the authors' clarifications, and can say that they have removed my doubts regarding these points.

I only have a few remaining comments that I will list in the following.

1. lines 205-215: You conducted an analysis to predict tree size from the tree motion features and tree type and found that the factor "tree type" was the 9th most explanatory feature in the model of height and 6th in the model of dbh. What were the 1-8th/1-5th most explanatory features? Did you include tree age in your model? Why/why not? (see also my comment further below)

2. line 263: "The forest conifers covered the largest area..." - do you mean forest broadleaves?

3. line 278: The x-axis provides the best separation?

4. lines 325-333: is the wind environment affected by properties of the trees? e.g., might a canopy of a number of different tree types, different heights, etc. induce more turbulence than a rather homogenous forest?

5. lines 360-365: This also refers to the point above. If I understand correctly, wind turbulence may be influenced by the structure of the underlying canopy. (How) does this potential inter-dependency affect your analysis?

6. lines 385-387: Address here the raised issue of noise at low wind speeds.

7. lines 390-395: Can you explain why tree age is not included? Because it is correlated with tree height/size? I would expect that wind damage risk is increasing with increasing tree age.

8. General comment to future research directions: Are there any observations of deciduous and coniferous trees within the same forest available? This would (potentially) allow for a clearer study of differences between the tree types, as the trees would be exposed to more or less the same wind environment.

9. lines 438-440: "However, we could not accurately distinguish between the motion of open-grown and forest broadleaves, despite the substantial difference in tree shape between the extremes of this gradient." This sentence is unclear to me, could you rephrase it?

---

## Author Comment (AC2) · 14 Apr 2021

Thank you for your positive feedback and constructive review of our manuscript. Also, thanks for reading the other comments and our responses. This interactive review format is really helping the process. Please find our responses below:

1 Thanks for pointing this out. We mention the factor "tree type" to emphasise that it didn't explain much of the variation. I should also have included a reference to the supplementary materials here, since S4 contains a more detailed description of these models. I will expand the table in S4 to include the first 10 most important variables in each case. The short answer is that most of the important variables were from the catch22 feature set except for the fundamental frequency in the model for height, and the power spectrum slope in the model for DBH.

Tree age was not included because we didn't have this information for most of the trees in our study. Many of the trees are in natural forests or parks and we don't know their age. In some cases (e.g. Puerto Rico data set) it is very difficult to measure tree age due to the lack of distinct tree rings. However, this information may be useful for people using the data in future. I will ask the contributors to include it, where available, in the meta-data alongside the tree motion data deposit.

2 and 3 Yes – thank you for spotting these mistakes, we will correct them in the revised version.

4 and 5 Yes, the wind environment will depend on canopy structure so there will be differences between forest types and between forest and open-grown trees. This has implications for the clustering analysis (figure 3) but not for the changes over time (figure 4). Specifically, the clustering we observe is potentially due to both the similarities in tree motion between tree types as well as similarities in the wind environment.

We state this limitation in lines 275-277, 325-333 and 360-365. We will make this clearer by updating line 328 to: 'This regularity could be related to the wind environment (i.e. a turbulent wind environment over a rough forest canopy leading to lower regularity in wind loading) as well as the properties of the tree. Therefore, the observed clustering could be due to similarities in the wind environment as well as similarities in the tree response'.

6 We will add: 'At very low wind speeds the tree motion will be small and noise from the sensors may be significant. However, we do not believe that this trend is driven by noise since many of the sensors were extremely sensitive and the trend is similar across all data sets.'

7 We didn't have tree age data for the trees in this study. There is some interesting work showing that wind damage risk changes with age in a conifer plantation (increasing with age initially and then decreasing) but we could not conduct a similar analysis here. Incidentally, many of the forests are mixed species with very different age – size

relationships.

8 We are not aware of any data set like this, but it would be a good way to test it. It would be important to get a sufficient sample of different size trees in both types of course.

9 We will rephrase line 438 to 'However, we could not reliably distinguish open-grown broadleaf trees from forest broadleaves based on their motion in the wind.'

---

## Author Response (AR1)

This document has the reviewer comments in *blue italics* and our responses in normal text.

**Response to reviewer 1**

*The paper evaluates the factors influencing the tree responses to wind loading. To that purpose, the Authors resembled an heterogeneous database including tree motion and wind velocity time series over different trees: broadleaf and coniferous in forests and in open environments. Two factors influencing the tree responses emerged: the tree fundamental frequency ($f_0$) and the high-frequency slope of the tree power spectrum ($S_{freq}$). The Authors further found that (1) the fundamental frequency of forest conifers was better predicted according to the cantilever beam model while for broadleaves it was better approximated using a simple pendulum model, and (2) the slope of the tree energy spectrum remained constant from medium to high wind speeds. The paper concludes by some future research directions.*

*While I find that the Authors did a remarkable job resemble and analyzing this heterogeneous dataset, I find the results on the tree fundamental frequency not so new as compared to the first Author recent paper (Jackson et al. 2019), and I am quite skeptical about the meaning and importance given in the paper to the slope of the tree power spectrum and about the robustness of its evaluation.*

Thank you for your review. I think that some of the issues you raise, while appropriate for a simple single site study, are necessary complications when conducting a multi-site study such as this. For instance, some studies did not record wind data at all, others used a MET station near the forest to give hourly mean wind speeds and a few studies collected wind data at canopy height at 10 Hz using a 3D sonic anemometer. These differences preclude some types of analysis that would otherwise have been valuable. However, I think it is still useful to bring together these diverse data sets and compare them. Many of our conclusions (see 'Future work directions') address these differences between sites and we suggest best practice for future studies.

The aims of this paper were (1) to bring together numerous tree motion data sets; (2) to test whether previous results hold across multiple sites and (3) to describe the key similarities and differences that emerge when studying tree motion at this scale. I will respond to your points in order below.

*Major concerns:*
*a) In the Introduction section, the Authors give the impression that most of our understanding on the tree fundamental frequency comes from conifers and much less from broadleaves: "Previous data syntheses have focused on the fundamental sway frequency ($f_0$) of conifers [: : :]. This finding demonstrates that conifers can be approximated by a cantilever beam (i.e. a beam with distributed mass), but it is unclear whether this model extends to other types of trees." However, in a recent study (Jackson et al. 2019) the first author did apparently "a comprehensive view of natural sway frequencies in trees by compiling a dataset of field measurement spanning conifers and broadleaves, tropical and temperate forests" (their abstract). They concluded that "The field data show that a cantilever beam approximation adequately predicts the fundamental frequency of conifers, but not that of broadleaf trees" (still their abstract). I am, therefore, wondering what is new in this paper compared to Jackson et al. (2019) on the tree fundamental frequency? Does it require a new publication? What are the differences between the dataset used in both papers? The Authors should better position their paper compared to the previous one.*

As you point out, there is certainly some overlap between this paper and Jackson et al (2019). I agree with you that this analysis should be better positioned with respect to Jackson et al (2019) and the key differences explicitly stated. However, Jackson et al (2019) was primarily a simulation study, using finite element analysis to explore the relationship between tree architecture and natural sway frequencies, and it used the field data only to demonstrate the range of natural variation. Furthermore, there are three key differences between the analysis in Jackson et al (2019) and the current study:

(1) The data set we presented in Jackson et al (2019) was comprised of only summary data on the fundamental sway frequency, calculated using different methodologies in each individual study. In the current study, we have collated the raw tree motion data and calculated the fundamental sway frequency using the same method (wavelet analysis) for each time-series. This makes the current analysis more robust. I have added (Lines 82-83) 'Jackson et al 2019 demonstrated that tree architecture strongly influences $f_0$ using a combination of field data and finite element simulations.'

(2) There is more data in the current study. Comparing table S1 from Jackson et al 2019 with Table 1 of the current study shows 39 additional trees from conifer forests, 12 additional open-grown conifers, 62 additional open-grown broadleaves and 17 additional trees from broadleaf forests. The 62 additional open-grown trees, which were under-represented in Jackson et al (2019) make a substantial contribution to the current study.

(3) We did not test the simple pendulum model in Jackson et al (2019), which turns out to be quite important in the current study.

I have added (Lines 197-200) This study improves upon previous work (Jackson et al 2019) by (1) increasing the sample size in the underrepresented open-grown trees (2) calculating $f_0$ in a consistent way across the 243 timeseries instead of using summary data and (3) testing the simple pendulum model

I believe that, given the aims of this paper, it is valuable to include the fundamental frequency analysis (Figure 1) despite the overlap with Jackson et al (2019). This analysis gives the reader a good understanding of the variation in tree motion across our data set (i.e. the 'slowest' tree in our study takes approximately 10 seconds to complete one oscillation, while the 'fastest' takes approximately 0.7 seconds). It is also an important feature of tree motion that has been widely used in the literature and would therefore be a strange omission from a paper which aims to synthesis tree motion data.

I have broken down your major concern b) into three parts, which I will address in turn:

*b – part 1)*

*I find the meaning of the slope of the tree energy spectrum not clear in the paper. In lines 94-95, it is written that "the slope of the power spectrum ($S_{freq}$) can be used as an overall measure of energy transfer between wind and tree at different frequency ranges (van Emmerik et al., 2018; Van Emmerik et al., 2017)". I am not sure to agree with this statement that $S_{freq}$ represents the energy transfer between wind and tree. In my opinion, it is more representative of the tree energy transfer (cascading) or damping from $f_0$ to high frequencies.*

Thanks for raising this issue. We agree with you that $S_{freq}$ will be influenced by the damping and related to the stiffness of the tree. However, unlike typical building structures, the stiffness and

damping of the system will change with wind speed (particularly the aerodynamic damping), as the tree changes shape and the whole system deforms.

The spectral response of the tree is essentially the spectra of the wind modified by the tree response (close to a lumped mass damped harmonic oscillator for conifers). This is described in equation 27 in Gardiner 1992. This is also presented in Mayer (1987), Kerzenmacher and Gardiner (1998) and Sellier et al (2008). It seems that many of your criticisms stem from this difference in interpretation.

In order to make this point as clear as possible, we have added (Lines 100-103) 'The spectrum of the tree is essentially the spectra of the wind, modified by the properties of the tree (Gardiner, 1992; Kerzenmacher and Gardiner, 1998; Mayer, 1987; Sellier et al., 2008). Therefore, the slope of the tree spectrum ($S_{freq}$) is the result of the energy transfer between wind and tree as well as the energy transfer within the tree itself.'

We also added (Line 395: The spectrum of the tree (characterized by $S_{freq}$) is essentially the spectra of the wind, modified by the properties of the tree. )

*b – part 2)*

*Indeed, $f_0$ is usually located at the level of the inertial subrange of the wind velocity spectrum (see Figures S6 and S7), i.e. at frequencies larger than the frequencies of the main eddy motions at canopy top. I would think that the energy transfer between wind and tree occurs mainly at lower frequencies than $f_0$, where the tree power spectra exhibit the same distribution with frequency as the wind spectra. I think $S_{freq}$ reflects how the tree damps/transfers its energy independently of the wind. Maybe a way to verify which flow motions are involved in tree motions is to look at the momentum flux cospectrum, assuming that the momentum flux at canopy top is totally absorbed by the trees. For example, if you look at Figs 4 and 6 of Dupont et al. (2018, Agric Forest Meteorol., 262, 42-58), you can see that most of the canopy-top momentum flux occurs at frequencies lower than $f_0$. Smaller eddies than the dominant canopy-top eddies may transfer as well energy to the tree but I would think it mainly concerns branches and less the trunks where the measures presented in this paper have be done. Branch motions are not necessarily in phase with the trunk motions.*

The aim of our manuscript is not to identify which frequency ranges are most important, but to study the similarities / differences between trees. This helps contextualize the more detailed, single site studies. The fact that this study contains a number of diverse data sets (which is its strength) precludes the analysis you suggest, which would require high resolution wind data for all sites. I agree with you that the momentum flux cospectrum, or the transfer function would be preferable, but this was not possible for the majority of trees in this study due to the lack of high-resolution wind data.

All frequencies in the wind spectra necessarily stimulate tree motion, albeit rather unequally (this response is called the mechanical transfer function or admittance function). Previous studies have analysed which frequencies contribute most to this energy transfer (e.g. Dupont et al. 2018 and Gardiner 1995, Schindler 2008, Schindler and Mohr 2019). It is true that the dominant motion is triggered by the coherent turbulent structures in the wind (Schindler and Moher, 2019) but this does not mean there is no response at other frequencies. The wind drag will be primarily on the leaves / needles, resulting in their motion which will transfer to the stem. Motion also passes from the trunk back to the branches and to the leaves and is then dissipated as vortex shedding from the leaf tips (Spatz and Theckes, 2013). This leads to a spectral short-cut in the wind spectra inside the canopy (Finnigan, 2000). Therefore, although the spectral shape of the tree displacement does reflect this

energy transfer, it is mixed with the direct response to the wind at all frequencies. We observe, therefore, a superposition of tree fundamental mechanical response and a supplementary response due to the transfer of energy between different frequencies that overall leads to a transfer from low frequency motion to high frequency needle / leaf waving.

*b – part 3)*

*The lower $S_{freq}$ for broadleaves than for conifers may just reflect their difference in architecture. I would think that $S_{freq}$ is representative of the tree properties, but not representative of the wind. Is it really new/surprising to observe differences between tree species in energy cascading/damping knowing that this mechanism depends on the tree properties (architecture, stiffness: : :)?*

We agree with you that some (or even most) of the differences between trees arise from their different architectures. To our knowledge, no study has compared tree motion spectra across multiple species and different genera and growing conditions before. Our aim was not to produce surprising results, but to test whether or not different trees behaved in different ways. Perhaps the surprising part (and not what we expected initially) is the degree of similarity across such a diverse data set. $S_{freq}$ is rather consistent across all trees and decreases to a value of -3 at around 4 m/s in most cases. We do not speculate in the paper what this indicates but it might suggest a convergent evolutionary response to the danger of wind loading and an efficient method for dissipating energy.

*c) I am skeptical about the estimation of $S_{freq}$. First, tree energy spectra do not show scaling law between the fundamental frequency and high frequencies because of the presence of secondary maxima. It is therefore quite questionable to define a slope there. Second, this slope has been defined for a specific frequency range in Hz (lines 167-168), while this frequency range should start from a frequency depending on $f_0$. The estimated slope is certainly sensitive to the height and width of the tree spectrum fundamental and secondary maxima. There are many cases where it seems impossible to define $S_{freq}$ (see the tree spectra in Figures S6 and S7). I am, therefore, not surprise to see some erratic behaviors in $S_{freq}$ in Figures 4d. At least, these erratic behaviors should have been removed. Third, the Authors seem surprised and present as a result the fact that below a threshold wind speed value, $S_{freq}$ decreases with wind speed (Figures 4c-d). In my opinion, this decrease of $S_{freq}$ reflects the increasing noise of the tree data at high frequencies as the wind diminishes. With decreasing wind, the frequency of the main canopy motions gets lower. Consequently, $f_0$ is shifted to thebottom (high frequencies) of the inertial subrange of the wind velocity spectrum, where there is much less energy. The high-frequency trunk motions become negligible. I am, therefore, not surprise to see that $S_{freq}$ decreases with wind speed, its evaluation becomes irrelevant and should not be presented.*

I explored different methods to estimate the slope of the power spectrum in this study and found that using a large fixed interval was the most robust. This is similar to the approach taken in previous publications (van Emmerik et al 2017 and 2018). As you suggest, defining a different frequency range for each tree based on its fundamental frequency is an attractive idea, but in practice this method was too noisy to be applied across such a diverse range of data. A number of trees in our study exhibited no consistent fundamental frequency peak, so the automatically extracted slope of the power spectrum would be undefined in these cases. I have added (Lines 183-184) 'We tested calculating $S_{freq}$ with reference to the tree's $f_0$, but this method proved too noisy at low wind speeds.'

The definition we use gives a measure of the decline in energy in the tree spectra from 0.05 to 2 Hz. We do not have a good explanation for this change and do not present it as a key result (Lines 405-

408). However, we do not think that this is simply due to a declining signal-to-noise ratio. Even at relatively low wind speeds, many trees have large motions orders of magnitude larger than the sensitivity of the sensors. Obviously, our study contains a wide range of sensors and some of them may be noisy, but it also contains some extremely high-quality data sets which exhibit the same pattern. For example, accelerometers are extremely sensitive and the strain gauges are able to monitor the tiny diurnal fluctuations (few millimetres) in stem swelling as the trees stop respiring at night (Duperat et al, 2020).

I have added (Lines 328 - 330) 'However, we do not believe this pattern is mostly driven by noise since our data set contains a wide variety of sensors, some of which are extremely sensitive, and all of which all exhibit similar patterns'

*d) The location of the wind speed measurement should be clarified. The Authors compared the tree inclination angle against the wind speed between summer and winter but they do no say clearly where the wind has been measured (at canopy top, outside the plot: : :). It is just written that "We note that wind speeds were measured outside the forest or at canopy height in a single location" (Lines 304-305). This difference in wind speed measurements between experiments makes it difficult any comparison. For measurements outside the forest, are winter and summer measurements representative of the same footprint? It is difficult to conclude on Fig 4a because we do not know where the wind speed has been measured. The wind speed should be normalized by a reference wind speed.*

The location of the wind speed measurement is different for each study (Line 131), this is one of the challenges in working with such diverse data sets. Importantly, Figure 4d shows the changes over time for each tree individually, we are not comparing the magnitude of wind speeds across sites. It is these changes with increasing wind speed that we compare and find to be remarkably similar. I have updated this section to read (Lines 319-322) 'We note that wind speeds were measured in different locations for each experiment, mostly outside the forest or at canopy height, meaning that the reported wind speeds do not represent inflow wind speed for each tree.'

Part of the value of this study is that, in bringing together these data sets, we can compare the advantages and disadvantages of different experimental setups. This is discussed in the 'Future research directions' section. I have included more information on the location of the wind sensor for each site individually on the online data repository.

RE figure 4a – in this case a single tree is presented as an example and the wind speed measurement was taken outside the forest at a nearby MET station. I have included this information in the figure caption.

I have also added (Lines 423:425) 'To aid comparison between sites, locally measured wind speeds could be related to some near-surface reference wind speed available at hourly resolution, such as the hourly 10 m wind gust product provided by the European Centre for Medium-Range Weather Forecasts. '

*Specific comments:*

1) *Line 87: Which balance are you talking about? Can you be more specific?*

Thank you for pointing this out. I have updated the text to read (Line 91) 'The balance between energy transfer from the wind to the tree and energy dissipation by damping determines a tree's risk of wind damage'

> 2) Lines 97 and 46: "This study brings together all available data on tree motion". Thisis quite a strong statement. I see at least two datasets that were not considered or mentioned in this study: Sellier et al. (2008, Forestry 81, 279–297) and more recently Dupont et al. (2018, Agric Forest Meteorol., 262, 42-58)

Thanks for pointing this out, I have deleted the words 'all available' in all instances. I reached out to both of the authors you mentioned but received no data. I did receive data from two additional studies (one open-grown broadleaf and four open-grown conifers). These data have been added to the analysis but they did not substantially change the results.

> 3) Line 102: Correct the parentheses for the Lubba et al. reference.

Thanks, I have corrected this.

> 4) Line 124: "We therefore focus on analyses which do not require these data (although we explore this data in supplementary S2)." This is confusing, why talking about these data if you did not use them?

This refers to the fact that different data sets have different wind data associated with them (L121-125). This is a key point in understanding the challenges of this type of multi-site study, as discussed in detail above. I mention these high-resolution wind data because they are the 'gold standard' in experimental design and many of the more recent papers on this subject rely on these data.

> 5) Lines 138-139: "although open-grown trees exposed to strong winds may experience slowly varying displacements due to the mean wind speed on this timescale (Angelou et al., 2019; James et al., 2006)." I do not understand. Could you clarify?

When processing raw tree motion data, it is generally necessary to define a zero or mid-point at which the tree is vertical. This allows the data to be interpreted in terms of displacement from this position in different directions. This is complicated by the fact that some motion sensors 'drift' over time, i.e. an offset builds up due to a number of factors (L133-134). Most previous studies have done this by assuming that the tree will pass through its mid-point often. Therefore, we subtract a running mode or apply a high-pass filter to each interval of data (1 hour or 10 minutes are commonly used) to correct for this offset. This has been shown to be effective in a number of studies and is standard practice for trees in forests. However, open-grown trees may behave differently. In particular, they may be displaced from the vertical for a long period of time due to the effect of the mean wind speed. This is discussed in detail in the cited paper (Angelou et al 2019) and we mention it here as a caveat to our results. I have updated lines 145-149 to read 'This period seems reasonable to capture most effects of wind-tree interaction, although open-grown trees exposed to strong winds may be displaced from their resting position for continuously for long periods of time due to the mean wind speed on this timescale (Angelou et al., 2019; James et al., 2006). Therefore, different data filtering techniques may be more appropriate for open-grown trees (Angelou et al., 2019).'

I have updated the text (Line 156) to read 'We selected 1-hour samples during the windiest conditions available for each tree'

7) *Lines 227-229: "The frequency range in which energy transfers from the wind to the tree will therefore shift, and this will be reflected in the slope of the power spectrum." I do not understand why it will be reflected in the slope of the power spectrum. Shift in which direction?*

The tree spectrum is essentially the wind spectrum modified by the tree response. $S_{freq}$ is the slope of the tree spectrum. If either the wind spectrum or the tree response changes (due to increased wind speed or streamlining, respectively) we expect $S_{freq}$ to change. As for the direction, an increasing wind speed should lead to more energy at higher frequencies.

As discussed above, the tree spectrum is essentially the wind spectrum modified by the tree response. We state explicitly that $S_{freq}$ will depend on the wind spectrum because different sites may have different spectra (i.e. a uniform conifer forests compared to a multi-layered tropical forests or an open-grown tree in a park).

9) *Lines 328-329: "This regularity could be related to the wind environment (i.e. a turbulent wind environment leading to lower regularity)". I do not understand the notion of "wind environment", nor what is written between parentheses.*

I have updated the text (Lines 345-346) to 'This regularity could be related to the wind conditions (i.e. a turbulent wind in a multi-layered forest leading to lower regularity) or the properties of the tree.'

For example, tree A, situated in a dense forest will experience turbulent wind conditions with most of the loading on the upper canopy, while the lower parts of the tree are sheltered. Tree B, growing on a flat coastline with no other trees or obstacles nearby will experience consistent wind speeds and the wind loading will be distributed across the height of the tree. In the parentheses I am suggesting that the motion of tree A may be more regular than that of tree B.

10) *Line 342: "separate models", to which models are you referring to?*

There have been a number of attempts to model tree response to wind loading (e.g. ForestGales) but these are mostly based on uniform stands of conifer plantations. It would be valuable if these models could be transferred to more 'natural' forest environments and to open-grown trees. If we had found a clear separation between different types of trees in Figure 3, i.e. they move in distinct ways, transferring these models between these types of trees would have been unlikely to work. We therefore suggest it is good news, from a modelling perspective, that the trees overlap in Figure 3. I have updated the text to read (Lines 359-361) 'This negative result may be interpreted as good news

from a wind damage modelling point of view, since it suggests that existing models (developed for conifer forests) may be generalisable to open-grown trees.'

> *11) Line 347: "which suggests a difference in the frequency range of the peak wind-tree energy transfer." Could you clarify? I do not understand.*

I have updated this to (Lines 364-365) 'This was driven by their steeper $S_{freq}$, which suggests a difference in either the wind-tree energy transfer or the damping mechanisms.'

> *12) Line 378: "This could be because the size of the turbulence structures containing most energy are smaller than the tree crown at high wind speeds". I do not think so. The main turbulent motions at canopy top should not change much size with wind speed. In my opinion, the plateau of $S_{freq}$ just shows that $S_{freq}$ does not inform on the wind-tree energy transfer but only or mainly on the energy cascading/dissipation of the tree motions from $f_0$ to high frequencies, which only depends on the tree properties and much less on the wind intensity.*

We presented this mechanism as a possible explanation for our observations, it was purely speculative and I have deleted it from the discussion.

> *13) Line 422: "the oscillatory component of tree sway diminished with wind speed for four forest Scots Pine trees". I am not sure to understand this sentence and how it demonstrates the presence of a resonance mechanism.*

This sentence is describing the work in the cited paper (Schindler and Mohr, 2018) and suggests that no resonant effects were found. They used singular spectrum analysis to separate the oscillatory components of tree sway and found that their importance diminished wind increasing wind speed. We are suggesting that this analysis should be carried out across our newly collated data set, in order to test whether this result holds more generally. I have added (Lines 445-449) 'This result shows that no resonance between wind and tree occurs at the observed wind speeds, and that resonance is unlikely to occur at higher wind speeds'.

> *14) Lines 423: "the all the trees", please rephrase.*

I have corrected it.

> *15) Line 447-448: "All trees in this study exhibited a remarkably constant slope of the power spectrum from medium to high wind speeds in both summer and winter. This suggests that the relationship between wind loading and tree deflection is simply related to wind speed in the high wind speed range." So, it does not depend on the tree properties? I would say that it shows that $S_{freq}$ depends on tree trunk and branches properties and less on the presence or not of leaves.*

The value of $S_{freq}$ does depend on tree properties, we can see this in our comparison across trees (Fig 2b). We argue that the lack of changes in $S_{freq}$ suggests there are no substantial changes in the tree response, such as additional damping mechanisms or resonant effects.

*16) Figures S6-S7: Is it the same graduations for the vertical axis of the tree and wind energy spectra? Could you show the graduations? It would be nice to show the -2/3 slope. Some velocity energy spectra look very flat but it may just be a question of graduation.*

Thanks for your suggestion, I have updated the supplementary figures accordingly.

**References**

Duperat, M., Gardiner, B., Ruel, J.-C., 2020. Testing an individual tree wind damage risk model in a naturally regenerated balsam fir stand: potential impact of thinning on the level of. For. An Int. J. For. Res. 1–10. https://doi.org/10.1093/forestry/cpaa023

Dupont, Sylvain, et al. "How stand tree motion impacts wind dynamics during windstorms." *Agricultural and Forest Meteorology* 262 (2018): 42-58.

Gardiner BA (1992) Mathematical modelling of the static and dynamic characteristics of plantation trees. In: Franke J, Roeder A (eds) Mathematical modelling of forest ecosystems. Sauerländer, Frankfurt/Main, pp 40–61

Gardiner, B. A. "The interactions of wind and tree movement in forest canopies." *Wind and trees* (1995): 41-59.

Kerzenmacher, Tobias, and Barry Gardiner. "A mathematical model to describe the dynamic response of a spruce tree to the wind." *Trees* 12.6 (1998): 385-394.

Mayer H (1987) Wind induced tree sways. Trees 1: 195–206

Schindler, D., 2008. Responses of Scots pine trees to dynamic wind loading. Agric. For. Meteorol. 148, 1733–1742. https://doi.org/10.1016/j.agrformet.2008.06.003

Schindler, D., Mohr, M., 2019. No resonant response of Scots pine trees to wind excitation. Agric. For. Meteorol. 265, 227–244. https://doi.org/10.1016/j.agrformet.2018.11.021

Sellier, Damien, Yves Brunet, and Thierry Fourcaud. "A numerical model of tree aerodynamic response to a turbulent airflow." *Forestry* 81.3 (2008): 279-297.

**Response to reviewer 2**

Thank you for your positive feedback and constructive review of our manuscript. Also, thanks for reading the other comments and our responses. This interactive review format is really helping the process. Please find our responses below:

*1. lines 205-215: You conducted an analysis to predict tree size from the tree motion features and tree type and found that the factor "tree type" was the 9th most explanatory feature in the model of height and 6th in the model of dbh. What were the 1-8th/1-5th most explanatory features? Did you include tree age in your model? Why/why not? (see also my comment further below)*

Thanks for pointing this out. We mention the factor "tree type" to emphasise that it didn't explain much of the variation. I have added a reference to S4, which contains a more detailed description of these models. I have expanded S4 to include the first 9 most important variables in each model. The short answer is that most of the important variables were from the catch22 feature set except for the fundamental frequency in the model for height, and the power spectrum slope in the model for DBH.

Tree age was not included because we didn't have this information for most of the trees in our study. Many of the trees are in natural forests or parks and we don't know their age. In some cases (e.g. Puerto Rico data set) it is very difficult to measure tree age due to the lack of distinct tree rings.

*2. line 263: "The forest conifers covered the largest area: : :" - do you mean forest broadleaves?*

*3. line 278: The x-axis provides the best separation?*

Thank you for spotting these mistakes, I have corrected them.

*4. lines 325-333: is the wind environment affected by properties of the trees? e.g., might a canopy of a number of different tree types, different heights, etc. induce more turbulence than a rather homogenous forest?*

*5. lines 360-365: This also refers to the point above. If I understand correctly, wind turbulence may be influenced by the structure of the underlying canopy. (How) does this potential inter-dependency affect your analysis?*

*4 and 5*
Yes, the wind environment will depend on canopy structure so there will be differences between forest types and between forest and open-grown trees. This has implications for the clustering analysis (figure 3) but not for the changes over time (figure 4). Specifically, the clustering we observe is potentially due to both the similarities in tree motion between tree types as well as similarities in the wind environment.

We state this limitation in lines 207-208 and explain the implications in 379-384. We have updated lines 346-347 to 'This regularity could be related to the wind conditions (i.e. a turbulent wind in a multi-layered forest leading to lower regularity) or the properties of the tree'

*6. lines 385-387: Address here the raised issue of noise at low wind speeds.*

We updated lines 328-332 to: 'Part of this trend is likely driven by noise at low wind speeds, since the sensors will not reliably measure very low-level tree motion. However, we do not believe this pattern is mostly driven by noise since our data set contains a wide variety of sensors, some of which are extremely sensitive, and all of which all exhibit similar patterns.'

*7. lines 390-395: Can you explain why tree age is not included? Because it is correlated with tree height/size? I would expect that wind damage risk is increasing with increasing tree age.*

We didn't have tree age data for the trees in this study. There is some interesting work showing that wind damage risk changes with age in a conifer plantation (increasing with age initially and then decreasing) but we could not conduct a similar analysis here. Incidentally, many of the forests are mixed species with very different age – size relationships.

*8. General comment to future research directions: Are there any observations of deciduous and coniferous trees within the same forest available? This would (potentially) allow for a clearer study of differences between the tree types, as the trees would be exposed to more or less the same wind environment.*

We are not aware of any data set like this, but it would be a good way to test it. It would be important to get a sufficient sample of different size trees in both types of course.

*9. lines 438-440: "However, we could not accurately distinguish between the motion of open-grown and forest broadleaves, despite the substantial difference in tree shape between the extremes of this gradient." This sentence is unclear to me, could you rephrase it?*

I have updated this to (Lines 466-468) 'However, we could not reliably distinguish between open-grown and forest broadleaves based on their motion in the wind, despite their substantial architectural differences'.